# Noncovalent synthesis of homo and hetero-architectures of supramolecular polymers via secondary nucleation

Srinu Kotha[1], Rahul Sahu [2], Aditya Chandrakant Yadav[1,3], Preeti Sharma[4], B. V. V. S. Pavan Kumar [4], Sandeep K. Reddy [2] ✉ & Kotagiri Venkata Rao [1] ✉

The synthesis of supramolecular polymers with controlled architecture is a grand challenge in supramolecular chemistry. Although living supramolecular polymerization via primary nucleation has been extensively studied for controlling the supramolecular polymerization of small molecules, the resulting supramolecular polymers have typically exhibited one-dimensional morphology. In this report, we present the synthesis of intriguing supramolecular polymer architectures through a secondary nucleation event, a mechanism well-established in protein aggregation and the crystallization of small molecules. To achieve this, we choose perylene diimide with 2-ethylhexyl chains at the imide position as they are capable of forming dormant monomers in solution. Activating these dormant monomers via mechanical stimuli and hetero-seeding using propoxyethyl perylene diimide seeds, secondary nucleation event takes over, leading to the formation of three-dimensional spherical spherulites and scarf-like supramolecular polymer heterostructures, respectively. Therefore, the results presented in this study propose a simple molecular design for synthesizing well-defined supramolecular polymer architectures via secondary nucleation.

In recent years, living supramolecular polymerization (LSP) has emerged as a promising tool for synthesizing complex supramolecular homo- and heterostructures, offering meticulous control over their shapes and dimensions[1–4]. The pioneering work in this field began with the research groups led by Winnik and Manners et al., who first reported the living crystallization-driven self-assembly (LCDSA)[5]. They used poly(ferrocenyldimethylsilane) cored block copolymers to synthesize various supramolecular nanoarchitectures such as one-dimensional (1D) cylinders[6–8], 2D platelets[9,10], dendritic micelles[11,12], rod-coil micelles[13], and 3D spherulites[14,15] with excellent precision to name a few. Despite the significant progress in the supramolecular polymerization of small molecules, realizing LSP was challenging as no rational design principles were available to create dormant monomers

or aggregates. To this end, Sugiyasu et al. demonstrated LSP in small molecules using hydrogen-bonded (H-bonded) porphyrins, which form metastable J-aggregates[16]. Shortly after, Aida et al. also achieved kinetically trapped monomers using H-bonded corannulenes via intramolecular H-bonding[17]. Both these trapped states are activated using thermodynamically stable SPs as seeds to form SPs with controlled lengths. These inspiring outcomes further led to synthesizing block SPs with controlled sequences[18–23].

Over the past few years, a few researchers have reported the phenomenon of molecular self-assembly operating at higher hierarchical levels, thereby modulating the topology of functional SPs through a secondary nucleation mechanism[24–29]. While this phenomenon has been comprehensively studied theoretically and

[1]Department of Chemistry, Indian Institute of Technology Hyderabad, Kandi, Sangareddy, Telangana 502284, India. [2]Centre for Computational and Data Science, Indian Institute of Technology Kharagpur, Kharagpur, West Bengal 721302, India. [3]Department of Materials Science and Metallurgical Engineering, Indian Institute of Technology Hyderabad, Kandi, Sangareddy, Telangana 502284, India. [4]Department of Chemistry, Indian Institute of Technology, Roorkee 247667 Uttarakhand, India. ✉e-mail: skreddy@iitkgp.ac.in; kvrao@chy.iith.ac.in

experimentally in the context of amyloid self-assembly[30–33], it remains a relatively unexplored frontier within the domain of functional SPs. Recently, Yagai et al. have reported the creation of molecularly interlocked nanostructures with a remarkably high level of organization through the utilization of a secondary nucleation strategy[28]. Similarly, Sugiyasu et al. have described the formation of double-stranded Archimedean spirals through a secondary nucleation event[25]. George et al. have recently reported the emergence of secondary nucleation-mediated surface-anchored self-sorted SPs, secondary SPs, and the production of seed-induced, non-covalently cross-linked hydrogels[24,27,29]. Despite these interesting results, achieving fine control over the topology of functional SPs requires a much-needed understanding of the molecular structure, kinetics, and underlying mechanisms that govern secondary nucleation.

Currently, LSP and secondary nucleation pathways are explored in small molecule-based H-bonded π-systems mainly because of two reasons: (i) growth via the nucleation-elongation process and (ii) the ease of achieving kinetically trapped monomers/aggregates[16,17,34–43]. However, H-bonding interactions typically result in 1D morphology owing to the strong directionality of H-bonding functional groups[34–39]. Hence, the synthesis of complex supramolecular architectures using H-bonded π-systems is limited to a few sets of molecules[25,28,44,45]. On the other hand, Che et al. demonstrated LSP using non-H-bonded π-systems by activating their off-pathway aggregates via homo and hetero-seeding methods leading to the synthesis of interesting homo and block supramolecular architectures[46–48]. However, rational molecular design principles to realize LSP and secondary nucleation in other types of π-systems that form SPs without the aid from H-bonding

interactions need to be formulated for further exploration. In this context, we have recently reported that non-H-bonded simple π-systems such as alkyl perylene diimides (PDIs) show cooperative supramolecular polymerization guided solely by dispersive interactions with cooperative factors on par with H-bonded π-systems[49]. Hence, exploring alkyl PDIs to realize LSP and secondary nucleation pathways can broaden the scope of small molecules and expand the non-covalent synthesis toolbox to create complex supramolecular architectures. However, the generation of dormant monomers or off-pathway aggregates from alkyl PDIs must be addressed.

Here, we present a noticeable improvement in the field of SPs by expanding the structural diversity of monomers that display LSP and secondary nucleation to synthesize SPs with interesting topologies. For this purpose, we have selected a simple π-system, PDI substituted with 2-ethylhexyl side chains (**2EH-PDI**), as it is known to form SPs via nucleation-elongation mechanism (Fig. 1a)[49]. Despite the lack of any H-bonding functional groups, **2EH-PDI** forms dormant monomers in solution from the monomeric states upon rapid cooling due to the existence of multiple conformations at high temperatures. By optimizing the solvent conditions, we successfully stabilized the **2EH-PDI** molecules as dormant monomers in solution at room temperature (Fig. 1b). Next, these dormant monomers were converted into complex supramolecular architectures via seed-induced LSP and secondary nucleation. To accomplish this, we have employed three distinct seeding methods: (i) homo-seeding, (ii) shear forces, and (iii) hetero-seeding (Fig. 1b, c). By adding various mol% of prefabricated **2EH-PDI** SP seeds (homo-seeding), we could synthesize 1D SPs with controlled lengths, with the primary nucleation-elongation event taking

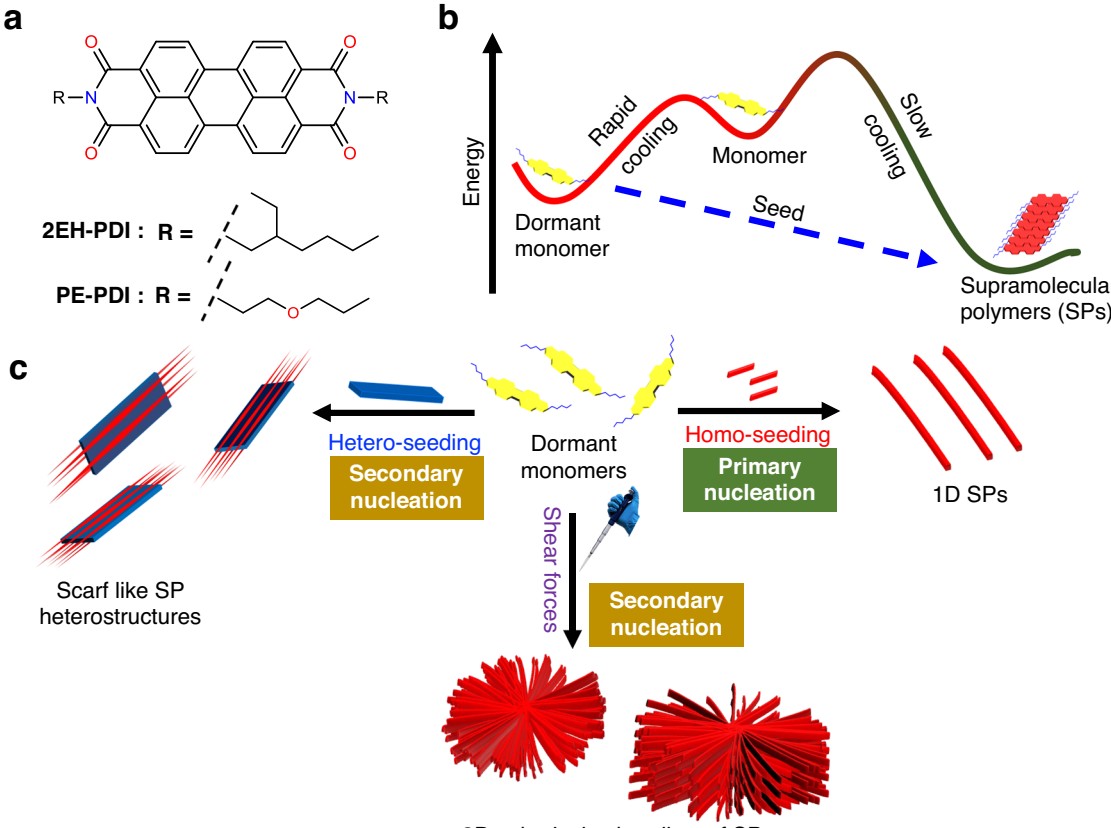

**Fig. 1 | Molecular structures and summary of various seeding methods.**
**a** Chemical structures of **2EH-PDI** and **PE-PDI**. **b** Schematic representation of the formation of dormant monomers by **2EH-PDI** and their transformation to thermodynamically stable supramolecular polymers (SPs) via seeded living supramolecular polymerization. **c** Schematic representation providing an overview of three

non-covalent synthesis approaches, demonstrating how the activation of dormant monomers of **2EH-PDI** leads to the formation of 1D SPs via homo-seeding, 3D spherical spherulites via shear forces and scarf-like SP heterostructures via hetero-seeding.

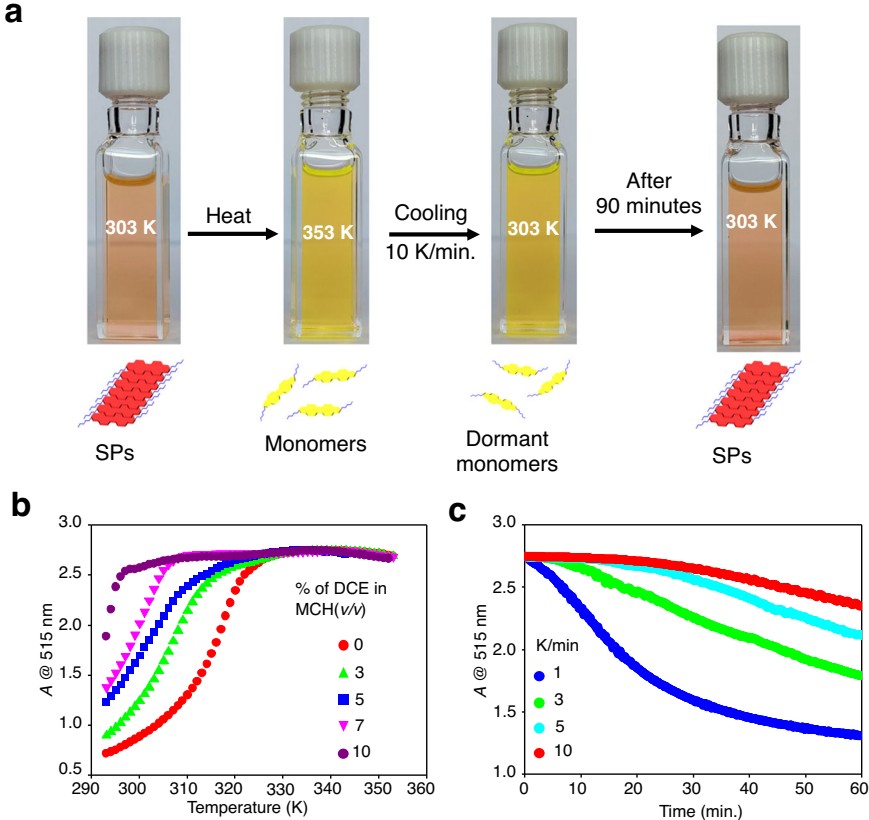

**Fig. 2 | Synthesis of dormant monomers. a** Synthesis of dormant monomers of **2EH-PDI**. Heating the supramolecular polymers (SPs) formed at 303 K to 353 K leads to the formation of monomers in MCH*. Later, by rapid cooling (10 K/min.) to 303 K led to the formation of dormant monomers. When these dormant monomers were left undisturbed at 303 K, SPs were formed over a period of 90 min.

**b** Temperature-dependent UV-vis absorption variation of **2EH-PDI** monitored at monomer wavelength (515 nm) at various volume percentages of DCE in MCH (*c* = 50 μM). **c** Time-dependent UV-vis absorption variation of the **2EH-PDI** at 515 nm at various cooling rates in MCH* (*c* = 50 μM) at 303 K.

precedence. Interestingly, when we mechanically agitated the dormant **2EH-PDI** monomers via shear forces (stirring or pipetting) in a controlled manner, inducing seed formation within the solution, we observed the formation of SPs with 3D spherical spherulite architectures[50], where the secondary nucleation-elongation promoted the growth of SPs around the seed. While stir-induced secondary nucleation is commonly observed in amyloid systems[51] and the crystallization of inorganic crystals[52,53], it is a relatively rare phenomenon in functional SPs. Similarly, for hetero-seeding experiments, we have utilized 2D platelets formed by PDI substituted with propoxyethyl chains (**PE-PDI**) (Fig. 1a). In this case, the secondary nucleation event promotes the growth of 1D SPs of **2EH-PDI** above and below the surface of 2D platelets of **PE-PDI** across their length, resulting in scarf-like SP heterostructures (Fig. 1c). Such a controlled fabrication of SP architectures using simple π-systems via secondary nucleation at an elevated hierarchical level represents an important achievement. Moreover, we also provided the direct visualization of the formation of SP heterostructures in solution using optical microscopy.

## Results

### Formation of dormant monomers

Previously, we reported that **2EH-PDI** forms SPs in methylcyclohexane (MCH) and dimethyl sulfoxide (DMSO) via a cooperative mechanism[49]. **2EH-PDI** (50 μM) is molecularly soluble in chlorinated solvents such as chloroform ($CHCl_3$) and dichloroethane (DCE), as evidenced by UV-vis absorption spectra which shows characteristic monomeric peaks at 490 nm and 515 nm (Supplementary Fig. 1). In MCH, **2EH-PDI** (50 μM) forms SPs, as evidenced by relatively broad absorption spectra and a new redshifted band at 575 nm (Supplementary Fig. 1). These SPs could

be reversibly transformed into monomers at high temperatures (~363 K), as evident from the UV-vis spectra (Supplementary Fig. 2a). We have studied the supramolecular polymerization and depolymerization process of **2EH-PDI** by monitoring its degree of polymerization ($\alpha$) at 515 nm, ($\alpha$ @ 515 nm). Above the elongation temperatures ($T_e$), $\alpha$ is zero as **2EH-PDI** (50 μM) exists as monomers (Supplementary Fig. 2b). Interestingly, thermal hysteresis was observed in MCH during the supramolecular polymerization and depolymerization process. While heating (SPs → monomers), $T_e$ of **2EH-PDI** (50 μM) was observed at 353 K. However, in the cooling process (monomers → SPs), the critical elongation temperature ($T_e'$) was reduced to 330 K (Supplementary Fig. 2b). These results revealed the presence of pathway complexity, where the **2EH-PDI** monomers get trapped in a higher energy metastable state upon cooling the solution from monomers to form SPs (Figs. 1b and 2a). Probing the absorbance (A) at 515 nm (A @ 515 nm) indicates that by increasing the percentage of good solvent, such as DCE, the temperature region of the hysteresis loop enhanced (Fig. 2b and Supplementary Figs. 3–6). Monomers of **2EH-PDI** have a higher absorbance at 515 nm than its SPs. Hence, the decrease in the absorbance at 515 nm is an indication of supramolecular polymerization of **2EH-PDI** (Fig. 2b and Supplementary Figs. 3–6)[49]. At 10 vol% of DCE in MCH (MCH*), we obtained the kinetically trapped monomeric state (dormant monomer) at room temperature (Fig. 2 and Supplementary Fig. 6). Here, the cooling rate also dictates the stability of the dormant monomer. In MCH*, the dormant monomer of **2EH-PDI** obtained by the cooling at a rate of 10 K min$^{-1}$ is more stable (~ 25–30 min) at 303 K than cooling rates of 1, 3, and 5 K min$^{-1}$ (Fig. 2a, c).

In the case of H-bonded π-systems, intramolecular H-bonding is shown to be responsible for the formation of dormant

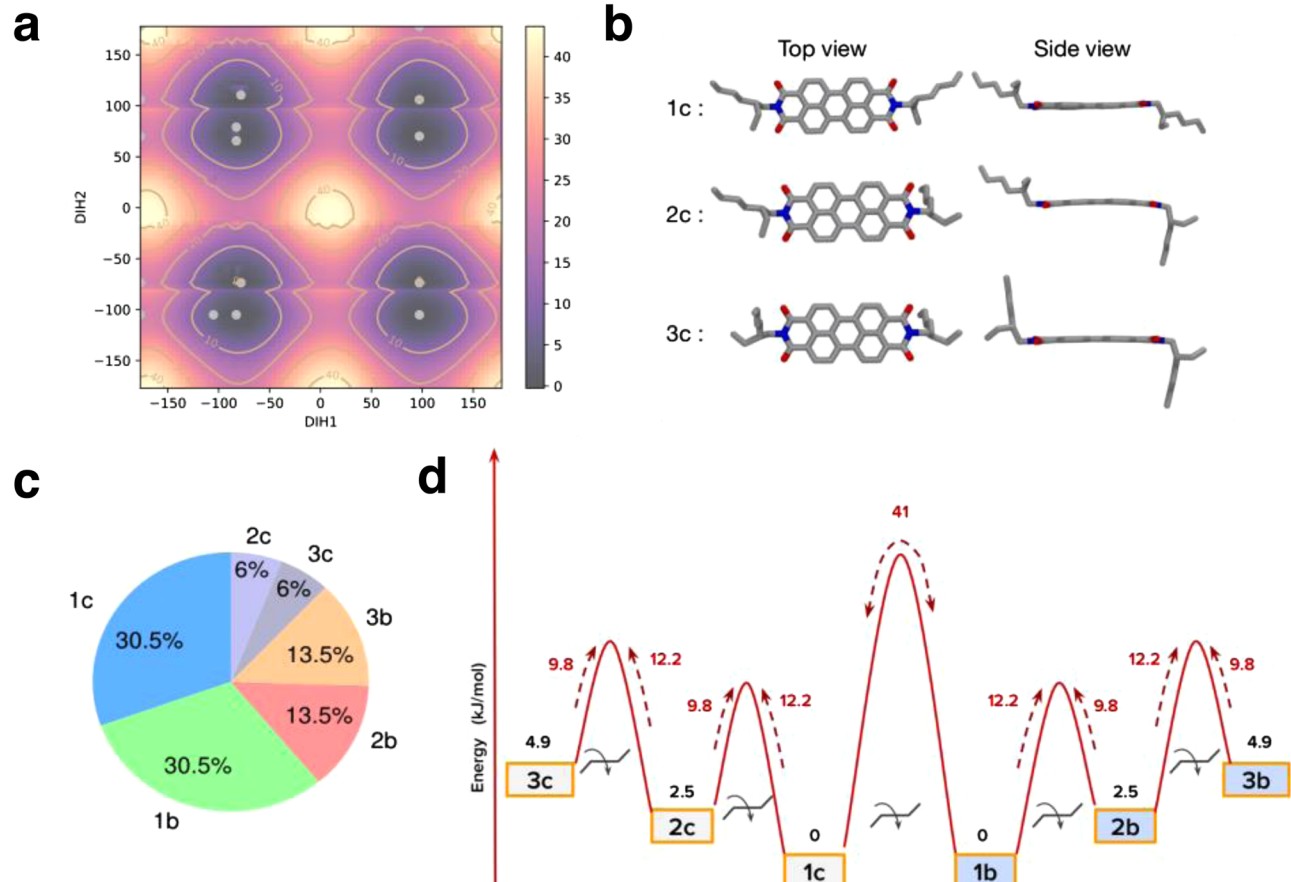

**Fig. 3 | Conformational analysis of 2EH-PDI. a** Potential energy surface of **2EH-PDI** molecule calculated using the semi-empirical PM6 method. The energy values, in kJ/mol, are shown on the color bar, and the two axes correspond to the two dihedral angles between the π-plane and the two side chains (C-N-C-C). All subsequent results were obtained using density functional theory. **b** Structure of the **2EH-PDI** molecule in three chair-type conformations labeled as 1c, 2c, and 3c. **c** Probability of finding each conformation at 353 K, calculated using the Boltzmann factor. **d** Relative energies (in KJ/mol) of the six conformers (shown in black) and the associated interconversion energy barriers (displayed in red), were both computed using density functional theory.

monomers[17,19,36,39]. However, in **2EH-PDI** it is surprising to see the formation of dormant monomers despite the lack of any H-bonding interactions. To gain insights into the kinetically trapped states of **2EH-PDI**, gas phase semi-empirical and quantum chemical calculations of the **2EH-PDI** monomer were performed using GAUSSIAN16[54]. Note that the reported energies are computed in the gas phase and will likely be altered in the presence of an explicit solvent. The details of the computational methods were provided in the Supplementary Information (Supplementary Note 2). Since all molecules exist in the monomeric state above the critical temperature, as a first step, we used quantum calculations to explore all possible conformations of the **2EH-PDI** molecule. We computed the two-dimensional potential energy surface (PES) by varying the two dihedral angles, each belonging to a side chain, defined between the π-plane and the side chains (C-N-C-C), using the semi-empirical PM6 method[55]. Fig. 3a depicts the PES of the **2EH-PDI** molecule, revealing several minima regions (highlighted as yellow colored dots). In the next step, we optimized each structure within the density function theory formalism using the ωB97xD/6-31 g(d) level of theory[56]. We identified six stable conformers of **2EH-PDI** with energy differences ranging from 0 to 5 kJ/mol. These conformations mainly fall into two types: chair and boat. In both cases, the alkyl moieties are positioned at angles of approximately 30 or 80° with respect to the π-plane. Figure 3b illustrates the three chair-type conformations (labeled as 1c, 2c, and 3c) arranged according to their stability. The analogous boat-type conformations (labeled as 1b, 2b, and 3b, respectively) are provided in Supplementary Information (Supplementary Fig. 7). Each boat conformer has the same

energy as its analogous chair conformation. The 1c conformer is the most stable among all chair-type conformers, with an energy lower by 2.5 kJ/mol and 4.9 kJ/mol compared to 2c and 3c, respectively.

The spatial arrangement of the 2-ethylhexyl side chains of 1b, 2b, 3b, 2c, and 3c conformations of **2EH-PDI** molecules constrains the π-π stacking interactions due to steric hindrance or defects. However, due to its favorable side chains spatial arrangement, the stack formed by the 1c conformer results in maximum π-π stacking interactions and lower steric hindrance, facilitating the formation of the stack[49]. This implies that the 1c conformer serves as the active species for the growth of a supramolecular polymer, and its higher concentration in the solution is necessary to observe stack formation. We calculated the probabilities of finding each conformation in the solution at 353 K using the Boltzmann factor, as shown in Fig. 3c. The results indicate that approximately 60% of **2EH-PDI** molecules adopt conformations 1c and 1b, while two-thirds of the remaining molecules are in conformations 2c and 2b. As a result, it is expected that in the solution, conformation 1c and 1b dominates, while other conformers are also significantly present. This disparity in the conformational population can lead to a decrease in the concentration of active monomers, resulting in the observed hysteresis during the cooling process. This finding aligns with the hypothesis proposed by Würthner and his team, who suggested that the kinetic inactivation of monomers depends on the energetic interplay between kinetically trapped, inactive species and active species for supramolecular polymerization[36].

To further explore the likelihood of transitions between these conformers, we conducted calculations to determine the energy

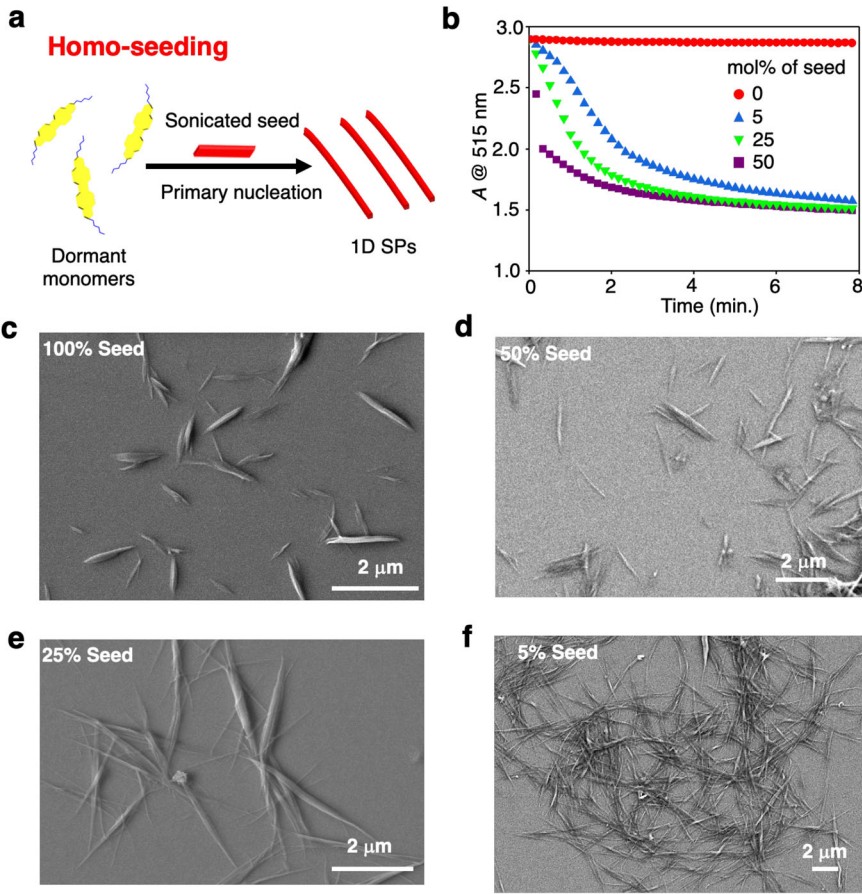

**Fig. 4 | Living supramolecular polymerization via homo-seeding. a** Schematic illustration of the homo-seeding approach of dormant monomers. **b** Time-dependent absorption changes observed for the **2EH-PDI** dormant monomers monitored at 515 nm after adding different mol% of seed. FE-SEM images of **2EH-PDI** obtained by spin coating the solutions on a silicon wafer **c** after sonication of prefabricated fibers, **d**, **e**, and **f** after performing LSP to **2EH-PDI** dormant monomers at varying [**2EH-PDI**$_{monomer}$] / [**2EH-PDI**$_{seed}$] ratios of 1:1, 3:1, and 19:1 respectively ($c$ = 50 μM, $l$ = 10 mm, solvent = MCH*). The images in **c**–**f** are representative of three experiments.

barriers involved, as depicted in Fig. 3d. The results reveal that the interconversion among either chair-type or boat-type conformations requires approximately 4 $k_B$T of energy, while the conversion from the 1c to 1b conformer requires approximately 14 $k_B$T of energy. These energy barriers are sufficiently high to hinder spontaneous nucleation, thereby governing the critical temperature during cooling. This observation is further supported by the dependence of the critical temperature on the cooling rates (Fig. 2c). Therefore, our computational findings suggest that thermal hysteresis primarily arises from the coexistence of multiple conformations at high temperatures. The elevated energy barriers inhibit the transition of kinetically trapped species into active species, which are essential for supramolecular polymerization.

**Homo-seeding**

The formation of dormant monomers by **2EH-PDI** (50 μM) in MCH* encouraged us to carry out LSP by adding the prefabricated SP seed of **2EH-PDI** (Fig. 4). When a DCE solution of **2EH-PDI** is injected into MCH, to result in the final solvent composition as MCH* (final concentration:50 μM), it spontaneously assembles into SPs of various lengths (Supplementary Fig. 8). Upon sonication of these long polydisperse SPs for 1 h at 303 K, they fragmented into many short seed-like fibers with active chain ends (Fig. 4c). UV-vis absorption and fluorescence spectra revealed that sonication did not alter the aggregation behavior **2EH-PDI** (Supplementary Fig. 9), and the average length of seed particles is around 700–800 nm (Fig. 4c). Seeded

growth polymerization was attempted immediately by adding the seed solution to the dormant monomers of **2EH-PDI**, and the final concentration of the solution was kept constant at 50 μM (Fig. 4a). As previously stated, the final length of SP always depends upon the mass ratio of added seed to monomer solution[16]. In this perspective, we have added different mol% of seed to dormant monomers and monitored through UV-vis spectroscopy and FE-SEM. The UV-vis absorption spectra of dormant monomers solution after adding the 5 mol% sonicated seed showed a sudden decrease in the monomeric absorption peaks at 515 nm and 480 nm, with an increase in the absorbance of the band at 575 nm (Fig. 4b and Supplementary Fig. 10). To further investigate the seed-indued living growth of **2EH-PDI** dormant monomers, we have added different mol% of seeds and monitored the kinetics of LSP (Fig. 4b). As expected, spontaneous elongation without any lag time was observed (Fig. 4b), and the length of resultant fibers analyzed by FE-SEM experiments has shown a systematic increase in the length of SPs with a decrease in seed mol%. From FE-SEM studies, we found that the average length of the fibers prepared from [**2EH-PDI**$_{monomer}$] / [**2EH-PDI**$_{seed}$] ratios of 19:1, 3:1, and 1:1 are 6 μm, 2 μm, and 1 μm, respectively, with the average seed length being 700–800 nm (Fig. 4d–f).

To obtain a clear understanding of the molecular mechanism, we conducted kinetic analyses at various concentrations of dormant monomer, specifically 40 μM, 45 μM, 50 μM, and 55 μM while keeping seed concentration constant at 12.5 μM. We monitored the growth kinetics by measuring changes in the absorbance at 480 nm

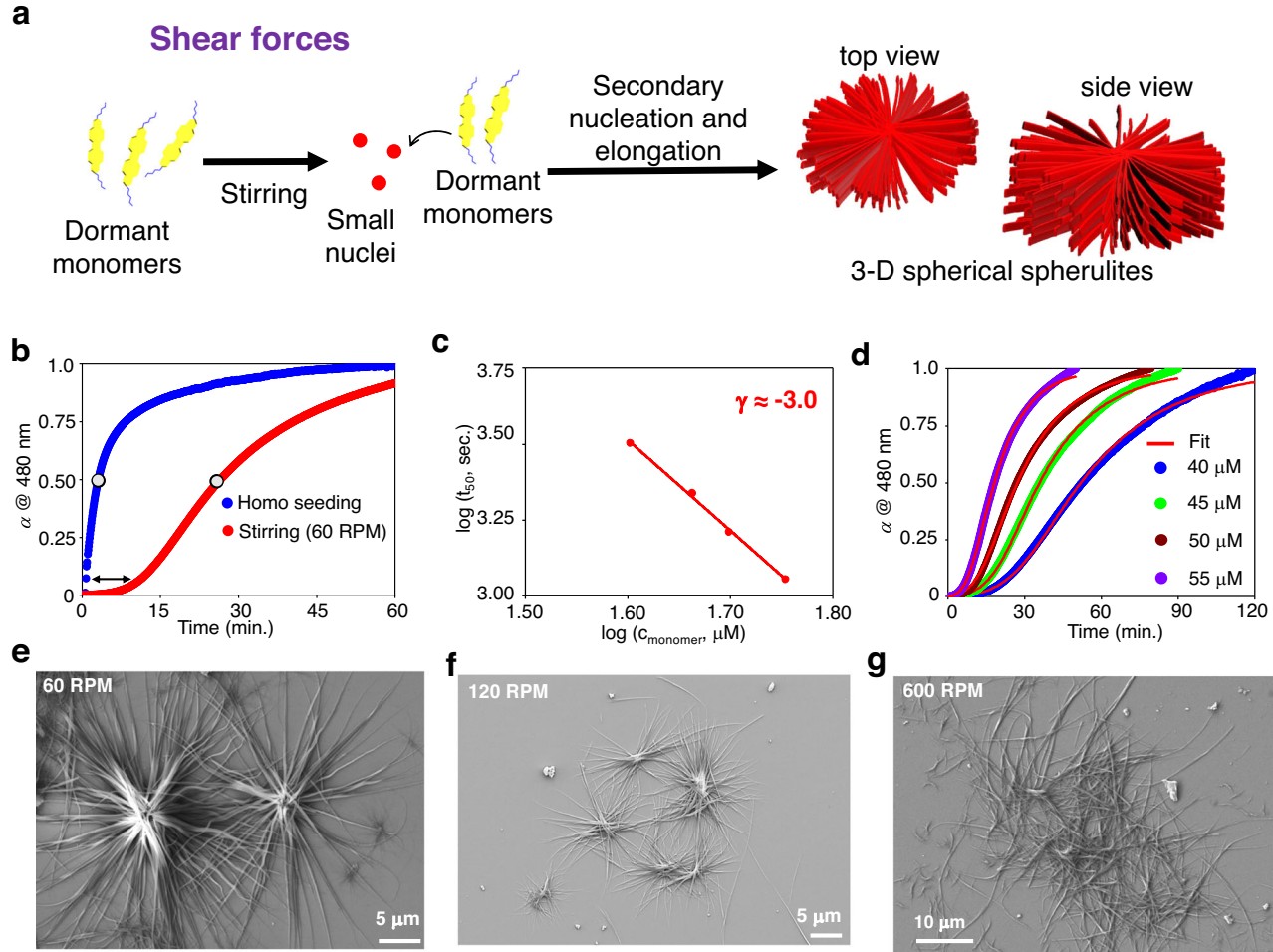

**Fig. 5 | Formation of 3D spherulites via stirring-induced secondary nucleation.**
**a** Schematic illustration of the formation of 3D spherical spherulites by **2EH-PDI** via secondary nucleation-elongation triggered by the application of shear forces (stirring) to its dormant monomers. **b** Illustration of differences between primary nucleation (homo-seeding) and secondary nucleation (stirring at 60 RPM) events. The sigmoidal growth of the **2EH-PDI** dormant monomer under stirring indicates the presence of the secondary nucleation process. The circles on the kinetic profiles indicate the half-time. **c** log-log plot of the half-times of stir-induced (60 RPM) supramolecular polymerization versus the original concentration of **2EH-PDI**. Symbols represent the experimental data; solid line is a power law fit. This fit shows a linear trend with a slope of −3.00 ± 0.37 referred to as the exponent coefficient (γ), indicating a monomer-dependent secondary nucleated supramolecular polymerization process. Since there is no seed is added externally, the concentration of the seed is zero. **d** Kinetic profiles of the concentration-dependent experiments at 60 RPM were obtained by monitoring the absorbance at 480 nm while varying the dormant monomer (**2EH-PDI**) concentrations to 40 μM, 45 μM, 50 μM, and 55 μM. FE-SEM images of **2EH-PDI**, obtained by spin coating the solutions on a silicon wafer after stirring the dormant monomers at **e** 60 RPM, **f** 120 RPM, and **g** 600 RPM for 60 min at 303 K (c = 50 μM, solvent = MCH*). The images in **e**–**g** are representative of three experiments.

($A$ @ 480 nm). The scaling exponent, which characterizes how the lag time of reaction or half-time ($t_{50}$) varies with the initial monomer concentration, was determined by the double-logarithmic plot of $t_{50}$ against monomer concentration, as depicted in Supplementary Fig. 11a under seeded conditions[24–30]. The obtained value for the scaling exponent is γ = −1.20 ± 0.09, corresponding to reaction order $n_1$ = 2.4. In the present case, we have not observed any fragmentation, and it is a monomer-dependent process hence γ can be a more negative value than −1[51,57]. The growth kinetics were modeled using a seed-induced nucleation-elongation framework from amyloid software (http://www.amylofit.ch.cam.ac.uk)[30] and fit well with the experimental data (Supplementary Note 3, Supplementary Fig. 11 and Supplementary Table 1). However, the same kinetic data showed a less favorable fit for the secondary nucleation-elongation model (Supplementary Fig. 12 and Supplementary Table 1). The above observations clearly indicate that the conversion of dormant monomers of **2EH-PDI** into SPs via homoseeding takes place through the primary nucleation-elongation mechanism.

## 3D spherical spherulites via secondary nucleation (shear-induced assembly)

Given prior publications on how mechanical agitation, such as stirring, aids the rate-determining nucleation process[35,37,58], we conducted time-dependent kinetic investigations of MCH* containing dormant monomers of **2EH-PDI** with stirring (Fig. 5). As expected, the nucleation process is expedited at 60 RPM, as seen by decreased monomeric absorbance after 10 min (Fig. 5b and Supplementary Fig. 13). Surprisingly, FE-SEM studies revealed that resultant self-assembled structures have 3D spherical spherulite-like structures (Fig. 5a, e and Supplementary Fig. 14)[14,15,50]. In most circumstances, the morphology obtained in LSP is the same as that obtained in thermodynamically controlled SPs. Although mechanical agitation, such as stirring or pipetting, speeds up the nucleation process, previous reports indicate that it has little effect on the morphology of SPs[37]. In the present case, mechanical agitation creates 3D spherical spherulites (*vide infra*) with micrometers length in all directions (Fig. 5e, f and Supplementary Fig. 14a–c). We conducted a

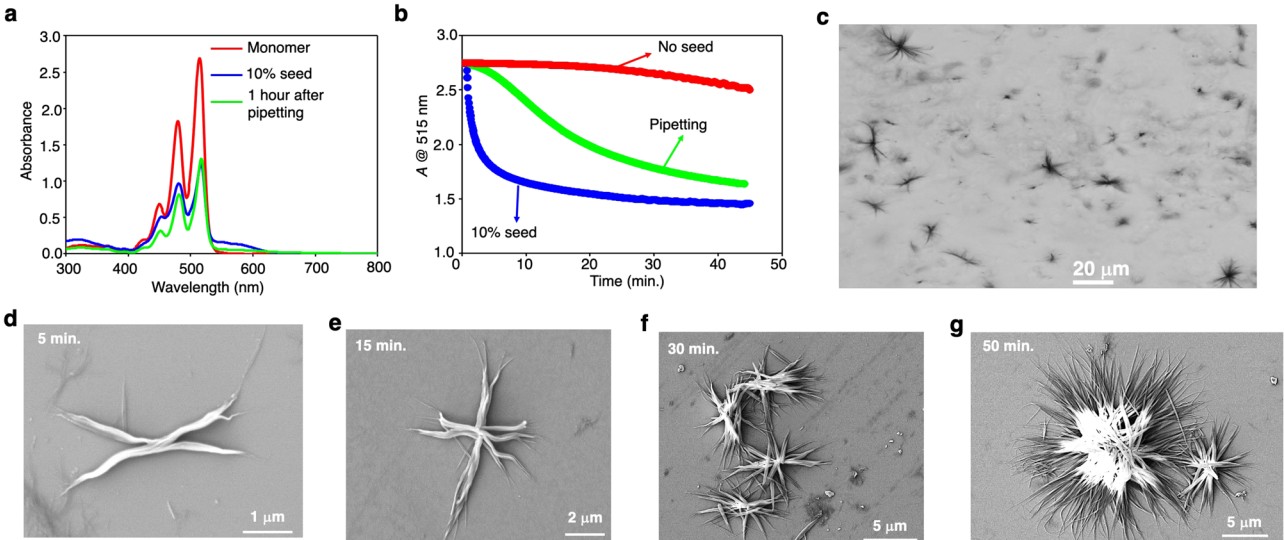

**Fig. 6 | Formation of 3D spherulites via pipetting-induced secondary nucleation. a** UV-vis absorption spectra of **2EH-PDI** dormant monomers (red), one hour after applying the slow mechanical agitation via repetitive pipetting (green), LSP after adding 10 mol% seed (blue) in MCH*. **b** Time-dependent variation in the absorbance of **2EH-PDI** at 515 nm after applying the mechanical agitation via repetitive pipetting to the dormant monomer (green). The control experiments by adding the 10 mol% seed (blue) and 0 mol% seed (red) to the dormant monomers of the **2EH-PDI** are shown for comparison. **c** Optical microscopy image of the spherulites obtained in MCH* after applying the slow mechanical agitation via repetitive pipetting. **d–g** FE-SEM images of **2EH-PDI**, obtained by spin coating the solutions on a silicon wafer after **d** 5 min., **e** 15 min., **f** 30 min. **g** 50 min. of applying the mechanical agitation via repetitive pipetting to the dormant monomers of **2EH-PDI** ($c = 50\,\mu M$, solvent MCH*). The image in **c** is representative of three experiments, images in **d–f** are representative of one experiment, and the image in **g** is representative of three experiments.

concentration-dependent kinetic analysis by monitoring the absorbance at 480 nm while maintaining a constant stirring rate of 60 RPM to understand the growth mechanism further. All the kinetic profiles exhibited a sigmoidal-like transition, which included a lag phase followed by an exponential phase (Fig. 5d). In stark contrast when considering homo-seeding, we observed spontaneous growth in kinetics, characterized by a non-sigmoidal nature and the absence of any lag phase (Fig. 5b). These characteristics are typical of the secondary nucleation-elongation process[24–30]. We determined the scaling exponent as $\gamma = -3.0 \pm 0.37$ (Fig. 5c), which is higher than the value obtained in homo-seeding experiments. This further indicates the presence of secondary nucleation events with a secondary reaction order of $n_2 = 5$. Furthermore, we comprehensively analyzed all the kinetics and found that all the data fit well with the unseeded secondary nucleation model, with an average mean squared error (MSE) value less than 0.0007 (Fig. 5d, Supplementary Note 3 and Supplementary Table 2), in contrast to the unseeded primary nucleation-elongation model, where the average mean squared error (MSE) value is more than 0.006. These kinetic analyses strongly suggest that the growth of 3D spherulites via shear forces takes place through a typical secondary nucleation-elongation mechanism.

Owing to their large 3D structures, the solutions having spherulites are inhomogeneous compared to the solutions having 1D SPs formed via primary nucleation (Supplementary Fig. 16a). We have also noticed that spherulites have better thermal stability than 1D SPs. A 50 μM solution of **2EH-PDI** SPs with spherulite morphology synthesized via secondary nucleation (stirring at 60 RPM) showed a melting temperature of 312 K, which is 7 K higher than the 1D SPs obtained via primary nucleation (with 5 mol% seed) (Supplementary Fig. 16b). Moreover when taken in 70% DCE in MCH at 303 K, 1D SPs are dissociated into monomers in less than 10 min, whereas 3D spherulites took more than 20 min (Supplementary Fig. 16c–e). All these observations suggest that the 3D spherulite structures of **2EH-PDI** formed via a secondary nucleation event are more robust than its 1D SPs formed via a primary nucleation event.

We noticed that up to 300 RPM, we could observe the formation of 3D spherical spherulites, and the size of spherulites decreased with increasing the RPM (Fig. 5e, f and Supplementary Fig. 14a–c). However, after stirring at 600 RPM, we could observe mostly 1D SPs and a few spherulites (Fig. 5g and Supplementary Fig. 14d). To understand the stability of spherulites at 600 RPM, we synthesized them at 60 RPM and stirred them at 600 RPM for 1 h. Interestingly, FE-SEM data reveal that these spherulites are stable even when the solution is subjected to strong mechanical agitation up to 600 RPM (Supplementary Fig. 17). However, when the dormant monomers of **2EH-PDI** were stirred at 600 RPM, mostly 1D SPs were formed. To understand this further, we conducted kinetic growth studies of **2EH-PDI** at 600 RPM across four different concentrations and obtained a scaling exponent ($\gamma$) close to −2.0 (Supplementary Fig. 18a). The primary nucleation reaction order ($n_c$) and secondary nucleation reaction order ($n_2$) were determined to be 4 and 3, respectively. Notably, the kinetic data showed a good fit for both the unseeded primary nucleation-elongation model (Supplementary Fig. 18) and the secondary nucleation-elongation model (Supplementary Fig. 19). This suggests that supramolecular polymerization at 600 RPM is driven by both primary and secondary nucleation events, resulting in the formation mixture of 1D fibers and 3D spherical spherulites (Supplementary Fig. 14d). This phenomenon was also observed in protein aggregation, particularly in amyloid formation, where both primary and secondary nucleation events occur[51,59,60].

We have also noticed that mechanical agitation via repetitive pipetting can also induce the formation of 3D spherical spherulites like stirring (Fig. 6 and Supplementary Fig. 20). To monitor the temporal evolution of the spherulites, we conducted a series of FE-SEM examinations at different intervals after applying weak mechanical stimuli such as pipetting. For this purpose, we have applied 5–6 times repetitive pipetting to the MCH* solution containing dormant monomers and left it undisturbed at 303 K. In this case, the assembly process began with a 5-min lag period, as indicated in Fig. 6a, b, and we observed the hedrite morphology after 5 min (Fig. 6d). The formation

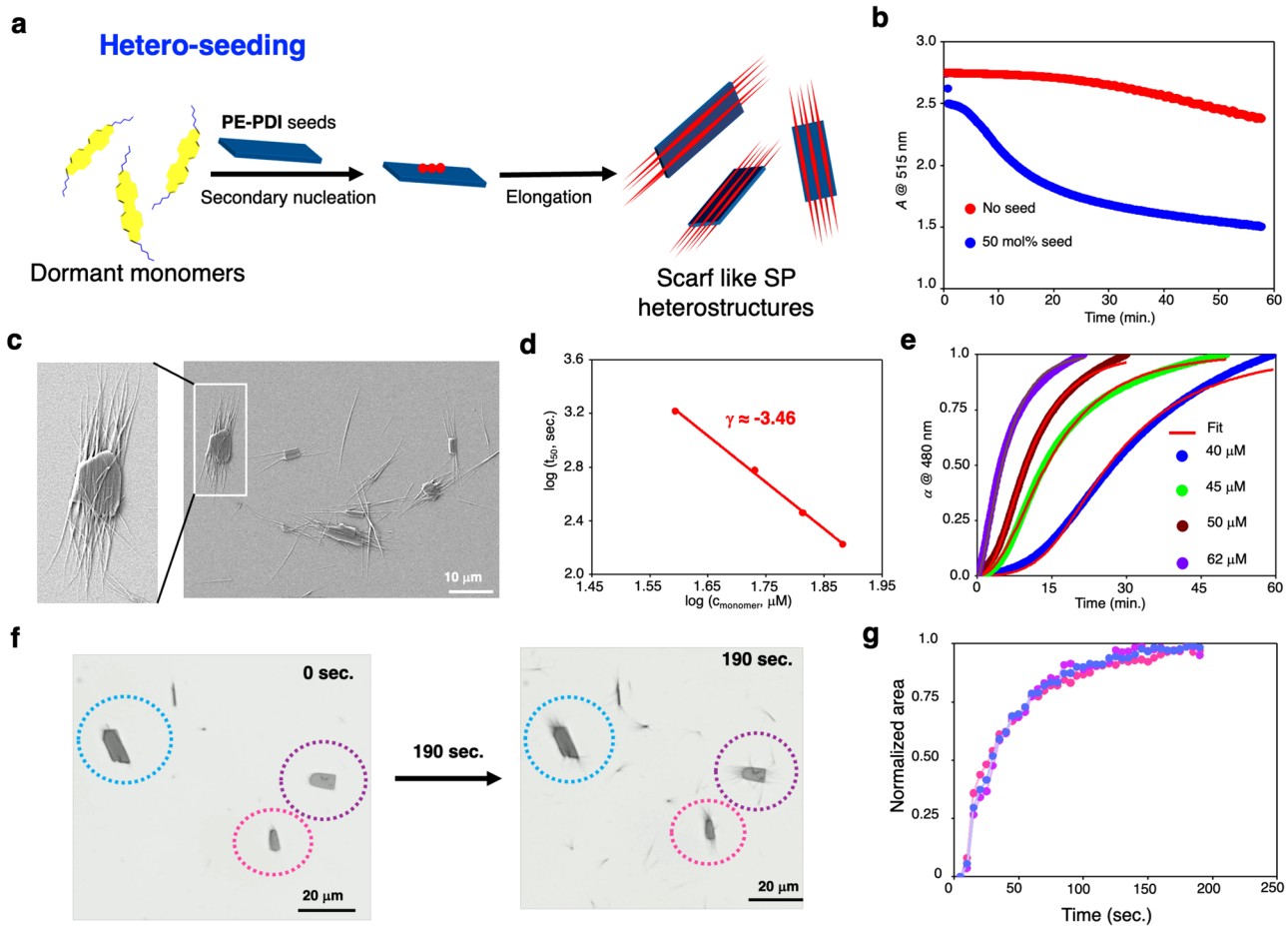

**Fig. 7 | Formation of scarf-like SP heterostructures via hetero-seeding-induced secondary nucleation. a** Schematic illustration for formation SP heterostructures obtained by adding **PE-PDI** 2D platelets to **2EH-PDI** dormant monomers via secondary nucleation. **b** Time-dependent variation in the absorbance of **2EH-PDI** dormant monomers at 515 nm after adding the 50 mol% of **PE-PDI** seeds (blue) and spontaneous polymerization of **2EH-PDI** (red) is shown for comparison. **c** FE-SEM image of SP heterostructures synthesized via hetero-seeding approach by adding 50 mol% **PE-PDI** 2D platelets seeds to **2EH-PDI** dormant monomers. **d** log-log plot of the half-times of hetero-seeding supramolecular polymerization versus the original concentration of **2EH-PDI**. Symbols represent the experimental data; solid line is a power law fit. This fit shows a linear trend with a slope of −3.46 ± 0.09 called exponent coefficient (γ), which suggests a monomer-dependent, seed-induced secondary nucleated supramolecular polymerization process. **e** Kinetic profiles of the concentration-dependent hetero-seeding experiments, obtained by monitoring the absorbance at 480 nm of **2EH-PDI**, at constant seed concentration (($PE-PDI)_{concentration}$: 25 μM) while varying the dormant monomer concentrations of **2EH-PDI** to 40 μM, 45 μM, 50 μM, and 62 μM. The sigmoidal growth of the **2EH-PDI** dormant monomers indicates a secondary nucleation-elongation process. **f** Bright-field microscopy images showing time-dependent growth of SP heterostructures after mixing the 50 mol% of **PE-PDI** seeds to the **2EH-PDI** dormant monomers in MCH* (encircled with dashed lines of blue, pink, and magenta). **g** Kinetics of growth of SP heterostructures from the **PE-PDI** seeds encircled in blue, pink and magenta dashed lines in panel (**f**) was tracked via image analysis to follow the increase in pixel area corresponding to the growth of new fibrils from the seeds. The image in **c** is representative of four experiments, and the images in **f** are representative of three experiments.

of intermediate X-shape (hedrite) morphology in the initial stages of growth is a characteristic of spherulite formation[50,61]. With the increase in time, we have observed fibril growth in all directions via small-angle branching[50,61], resulting in the formation of star-shaped morphology (Fig. 6e, f). We got larger 3D spherulite structures with increasing branches after 50 min (Fig. 6g and Supplementary Fig. 20b–d). The 3D structure of spherulite was further confirmed by FE-SEM images that provide a side view, which was obtained by tilting the substrate up to 70° (Supplementary Fig. 20c). To avoid any ambiguity on the formation of 3D spherulites in solution, the dormant monomers solution of **2EH-PDI** in MCH* was mechanically agitated via repetitive pipetting (5–6 times), and the resultant solution was observed under the optical microscope. Interestingly, after some time, we see the formation of 3D spherulite structures in solution (Fig. 6c). This unambiguously proves that the formation of 3D spherulites via slow mechanical agitation occurs in the solution.

The above results vividly illustrate the remarkable sensitivity of **2EH-PDI** supramolecular polymerization to even minor alterations in reaction conditions. The self-assembly behavior of **2EH-PDI** is primarily governed by the dominant primary nucleation event in the homo-seeding approach. However, by introducing shear forces through actions like pipetting and stirring, we could bias its self-assembly pathway, initiating secondary nucleation events that result in the formation of 3D spherical spherulite structures. In this context, the anticipated mechanism involves the agitation of dormant monomers, which facilitates generating a limited number of small nuclei (Fig. 5a). This, in turn, promotes secondary nucleation of high-energy dormant monomers on the small nuclei, followed by elongation in all directions, leading to the formation of low-energy SPs with spherulite architecture. Notably, stir-induced secondary nucleation events have been well-documented in processes such as crystallization[52,53] and protein aggregation[51]. However, it is worth mentioning that, to the best of our knowledge, no previous reports exist regarding their occurrence in functional SPs. Here, we demonstrated an important example where dormant **2EH-PDI** monomeric building blocks would lead to the formation of 3D spherulites via mechanical agitation, which is not the case in homo-seeding experiments.

## Supramolecular polymer heterostructures via secondary nucleation (Hetero-seeding)

The results obtained from homo-seeding and shear-induced assembly further encouraged us to explore the hetero-seeding approach (Fig. 7). It has already been demonstrated that supramolecular block copolymers can be synthesized using LSP via a hetero-seeding process, which can provide excellent structural and sequence control[10–13,18–22]. For the hetero-seeding approach, we prepared a **PE-PDI** having a propoxyethyl side chain at the imide positions (Fig. 1a). Like **2EH-PDI, PE-PDI** also self-assembles in MCH* via J-aggregation and forms 2D platelets (Supplementary Fig. 21 and further explanation there). **PE-PDI** seeds were prepared in MCH* via sonication for 1 h at 303 K and 50 mol% of these were added to the dormant monomers of **2EH-PDI**. We found a drop in **2EH-PDI** monomeric wavelength absorbances ($\lambda_{max}$ = 515 nm) after a 5-min lag period (Fig. 7a, b). FE-SEM images revealed that 1D SPs of **2EH-PDI** were grown on the surface of 2D platelets of **PE-PDI** and oriented across their length (Fig. 7c and Supplementary Fig. 22). Contrary to homo-seeding, which demonstrated spontaneous growth without any lag phase (Fig. 4b), the presence of a lag phase lasting for 5 min and the growth of **2EH-PDI** fibers on the surface of **PE-PDI** seeds suggests the presence of a secondary nucleation-elongation mechanism (Fig. 7b).

Like homo-seeding experiments, we conducted hetero-seeding experiments using various dormant monomer concentrations (40, 45, 50, and 62 μM) of **2EH-PDI**, while maintaining a constant seed concentration of 25 μM for **PE-PDI**. In all cases, the kinetic profiles exhibited a sigmoidal-like transition characterized by a lag phase followed by an exponential phase, which indicates the occurrence of a secondary nucleation event induced by seeding (Fig. 7e). As the monomer concentration increased from 40 μM to 62 μM, there was a noticeable decrease in the half-time. Plotting the half-time against the monomer concentration on a double-logarithmic scale yielded a scaling coefficient γ of −3.46 ± 0.09 (Fig. 7d), demonstrating a linear relationship and sign of a secondary nucleation event. Additionally, we derived a secondary nucleation reaction order of $n_2$ = 6 from this analysis. Furthermore, we conducted a comprehensive analysis of all the kinetics and found that they fit well with the secondary nucleation model, with an average mean squared error (MSE) value less than 0.0004 (Fig. 7e and Supplementary Table 4). These kinetic analyses strongly support the growth of SPs of **2EH-PDI** on the surface of **PE-PDI** seeds via a hetero-seeding approach triggered by a typical secondary nucleation process.

Since the PDI is common in both **2EH-PDI** and **PE-PDI**, we expected that SPs of **2EH-PDI** would grow from the ends of **PE-PDI**[20]. However, FE-SEM studies demonstrated that SPs of **2EH-PDI** are grafted on the surface of the **PE-PDI** nanoplatelets (Fig. 7c and Supplementary Fig. 22). To gain further insights into this phenomenon, we have conducted powder X-ray diffraction (PXRD) studies on SPs of **2EH-PDI, PE-PDI** and their heterostructures (Supplementary Fig. 23). PXRD revealed that 2D platelets of **PE-PDI** are more crystalline than **2EH-PDI** SPs. The SPs of **2EH-PDI** showed a broad and low intense peak at 2θ = 4.31° (d-spacing of 2.05 nm) indicating a poor crystalline order. On the other hand, the 2D platelets of **PE-PDI** showed good crystalline order, as evidenced by several sharp and high-intensity peaks. The first peak was observed at 2θ = 5.54°, corresponding to a d-spacing of 1.6 nm. Successive peaks at 0.8 nm (2θ = 11.09°) (d/2), 0.54 nm (2θ = 16.45°) (d/3), and 0.44 nm (2θ = 19.96°) (-d/4) are also observed. This indicates the lamellar-like packing of **PE-PDI** molecules with interdigitation of side chains in the 2D platelets (Supplementary Fig. 23). This observation further suggests that the side chain variation results in disparate crystal packing for both PDI derivatives. Due to these dissimilar lattice structures, **2EH-PDI** monomers do not grow at the chain ends of **PE-PDI** and prefer to grow on the surface of **PE-PDI** via secondary nucleation (Fig. 7a). Hence, the crystalline nature of individual structures is also retained in the heterostructures

(Supplementary Fig. 23). We have further controlled the grafting density of **2EH-PDI** SPs on the surface of **PE-PDI** 2D platelets by varying the mol% of **PE-PDI** (Supplementary Fig. 24). By decreasing the mol% of **PE-PDI** seeds from 50% to 30%, we have observed increased covering of the **PE-PDI** surface by the SPs of **2EH-PDI** compared to 50 mol % of the seed (Fig. 7c, and Supplementary Fig. 24a, b). By further decreasing the mol% of **PE-PDI** seeds to 15%, their surface is fully grafted by the SPs of **2EH-PDI** (Supplementary Fig. 24c, d).

We have also explored possibilities to provide the direct visualization of the growth of SPs of **2EH-PDI** on **PE-PDI** in solution via hetero-seeding (Fig. 7f, g). For this purpose, we have taken the MCH* solution containing dormant monomers of **2EH-PDI** and 50 mol% of **PE-PDI** seeds in rectangular long capillary cells. Initially, we observed only 2D platelets of **2EH-PDI** seeds in the solution. As time proceeds, we observe the emergence of SPs of **2EH-PDI** from **2EH-PDI** 2D platelet seeds across their length (Fig. 7f and Supplementary Movie 1). This is further supported by the increase in pixel area due to the growth of new fibrils from the seeds (Fig. 7g). Similar observations are made when these solutions are visualized under fluorescence mode (Supplementary Fig. 25 and Supplementary Movie 2). These observations unambiguously prove **PE-PDI** seeds induce the supramolecular polymerization of **2EH-PDI** dormant monomers via secondary nucleation in solution. Notably, such a direct visualization of the growth of SPs allows us to understand the temporal evolution of complex supramolecular structures from small molecular building blocks. The kinetics of growth probed through UV-vis experiments in 10 mm cuvette are slower than the optical microscopy experiments done in rectangular capillaries due to the mass transport limitations in narrow channels of capillaries (Fig. 7b, g). The growth of SPs of **2EH-PDI** from **PE-PDI** seeds within 200 μm rectangular capillaries followed by bright-field microscopy showed quick saturation of the growth within 3 min due to rapid depletion of local concentration of dormant monomers (Fig. 7g). On the other hand, in UV-vis absorption experiments performed in 10 mm cuvette, fresh monomers are continuously available to the growing SPs on the seeds. As a result, the process takes longer time (>20 min.) for the saturation.

Next, we have also explored the possibility of selectively disassembling one of the components from scarf-like 2D heterostructure. Interestingly, we found that SPs of **2EH-PDI** have less thermal stability than **PE-PDI** 2D platelets in MCH*. The SPs **2EH-PDI** showed melting at 307 K and significant depolymerization was observed at 323 K (Supplementary Fig. 26a, b). In contrast, **PE-PDI** seed particles are stable up to 330 K in MCH* (Supplementary Fig. 26a, c). Taking advantage of differences in thermal stability, the SP heterostructure solution was heated at 323 K for 15 min, and this hot solution was coated on the silicon wafer. FE-SEM images revealed the absence of SP heterostructure and the presence of segregated SPs of **2EH-PDI** and 2D platelets of **PE-PDI** (Supplementary Fig. 26d). This indicates that at 323 K, SPs of **2EH-PDI** are melted and detached from the surface of 2D platelets of **PE-PDI** and formed separately, probably during solvent evaporation. On the other hand, when this solution was cooled from 323 K to 303 K, and coated on a silicon wafer after waiting for an hour at 303 K, we observed the formation of SP heterostructures from FE-SEM images (Supplementary Fig. 26e, f). Furthermore, we immersed the silicon wafer containing SP heterostructures in MCH* solvent at 323 K for various time intervals and visualized them through FE-SEM, which showed only 2D platelet morphology (Supplementary Fig. 27b–d). This is due to the selective depolymerization and dissolution of SPs of **2EH-PDI** at 323 K from the surface of **PE-PDI** seeds into MCH* solution.

## Discussion

In this manuscript, we have presented a simple molecular design (**2EH-PDI**) to synthesize dormant monomers, which can be explored for architectural control of SPs via LSP and secondary nucleation. Theoretical studies indicate that dormant monomers of **2EH-PDI** are

formed due to the statistical distribution of various conformers of **2EH-PDI** monomers in the solution at high temperatures. We found the distinct nucleation events during the supramolecular polymerization of dormant monomers through homo-seeding, shear-induced assembly, and hetero-seeding, resulting in the creation of elegant supramolecular architectures such as 3D spherulites and scarf-like SP heterostructures. Interestingly, mechanical agitation of dormant monomers of **2EH-PDI** not only biases the mechanism of supramolecular polymerization from primary nucleation to secondary nucleation but also significantly alters the topology of SPs (1D SPs to 3D spherical spherulites). Moreover, the 3D spherulites obtained via secondary nucleation are more robust than corresponding 1D SPs obtained via primary nucleation. Low thermal stability SPs of **2EH-PDI** than 2D platelets of **PE-PDI** allowed us to remove and reattach the former ones from scarf-like SP heterostructures selectively. Our results further indicate that in addition to well-explored H-bonded π-systems for LSP and architectural control, simple π-systems such as alkylated PDIs would also be worth exploring in the same direction. This would be useful to expand the toolbox for non-covalent synthesis and provide access to complex supramolecular architectures.

## Methods

### Preparation of dormant monomers
A solution of **2EH-PDI** (0.3 mL) in DCE was injected into 2.7 mL of methylcyclohexane (MCH), promptly leading to the formation of SPs (Final concentration: 50 µM, final solvent composition: 10% DCE in MCH). Subsequently, this SPs solution was heated to 353 K, resulting in the formation of a monomeric state in the solution. Upon cooling this hot monomeric solution to 303 K at a rate of 10 K/min, kinetically trapped dormant monomers were formed and exhibited stability for over 30 min. Utilizing these kinetically trapped dormant monomers, three distinct seeding experiments were conducted, as outlined below.

### Homo-seeding
In this experiment, prefabricated SPs of **2EH-PDI** in 10% DCE in MCH were sonicated for 1 h at 303 K. To synthesize SPs with controlled lengths, different mol% of seed was introduced into the dormant monomers of **2EH-PDI** to result in a final concentration of 50 µM. For kinetic analysis, the seed concentration was fixed at 12.5 µM, and the dormant monomers concentration varied to 40, 45, 50, and 55 µM at 303 K.

### Shear-induced assembly
Following the formation of dormant monomers of **2EH-PDI** (40, 45, 50, 55, and 60 µM) at 303 K in 10% DCE in MCH, the solution was agitated at various RPM levels (60, 120, 300, and 600 RPM). We have also used reparative pipetting (five to six times) of dormant monomers **2EH-PDI** in 10% DCE in MCH for shear-induced assembly experiments.

### Hetero-seeding
Initially, **PE-PDI** SPs were generated by adding a DCE solution of **PE-PDI** 0.3 mL to 2.7 mL of MCH in a vial, resulting in the formation of 2D platelets of **PE-PDI** (Final concentration: 125 µM, final solvent composition: 10% DCE in MCH). These 2D platelets were sonicated for one hour at 303 K. Subsequently, this seed solution (20 vol%) was added to the dormant monomers of **2EH-PDI** (40, 45, 50, and 62 µM) at a temperature of 303 K.

## Data availability
The authors declare that the data supporting the findings of this study are available within the paper and its Supplementary Information files. All other information is available from the corresponding author upon request. Source data (https://zenodo.org/records/10866038) are provided with this paper.

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

## Acknowledgements

S.K.R. and K.V.R. acknowledge the Core Research Grant (SERB-CRG) from the Science and Engineering Research Board (CRG/2022/006457) for financial support. S.K.R. acknowledges the ISIRD award, IIT Khar-agpur for the financial support and the National Supercomputing Mis-sion (NSM) for providing computing resources of PARAM Shakti at IIT Kharagpur, which is implemented by C-DAC and supported by the Ministry of Electronics and Information Technology (MeitY) and Depart-ment of Science and Technology (DST), Government of India. B.V.V.S.P.K. the Core Research Grant (SERB-CRG) from the Science and Engineering Research Board (CRG/2020/006366) for financial support. S.K. thanks the CSIR, India, for the Senior Research Fellowship. R.S. thanks the UGC, India, for the Senior Research Fellowship. A.C.Y. thanks the MOE, India, for the Prime Minister research fellowship. P.S. thanks the UGC, India, for the Junior Research Fellowship. We thank Talluri Manoj from the Department of Materials Science and Metallurgical

Engineering, Indian Institute of Technology Hyderabad, for his help in FE-SEM measurements.

## Author contributions
S.K. carried out experimental work, and R.S. performed theoretical studies. K.V.R. conceived the idea, designed the experiments, and directed the project. S.K.R. directed the theoretical studies. A.C.Y. performed all the FE-SEM measurements. B.V.V.S.P.K. and P.S. conducted the optical microscopy studies. K.V.R., S.K.R., S.K. and R.S. wrote the manuscript.

## Competing interests
The authors declare no competing interests.
