## [Peer Review File · Nature Communications]

Noncovalent synthesis of homo and hetero-architectures of supramolecular polymers via secondary nucleationREVIEWER COMMENTS

Reviewer #1 (Remarks to the Author):

2023-NCOMMS-23-48849-Nature.Commun-Supramolecular.polymerization

Journal: Nature Communication (ID 2023-NCOMMS-23-48849)

Title: " Noncovalent Synthesis of Homo and Hetero-Architectures of Supramolecular Polymers via Secondary Nucleation "

Authors: Srinu Kotha,^a Rahul Sahu,^b Aditya Chandrakant Yadav,^{a,c} Preeti Sharma,^d B. V. V. S. Pavan Kumar,^d Sandeep K. Reddy,^{b,*} Kotagiri Venkata Rao,^{*}

In this manuscript, the authors report studies of living supramolecular polymerization of two perylene diimide derivatives. The authors had previously shown that bis-2-ethylhexyl perylene diimide (2EH-PDI) undergoes cooperative supramolecular polymerization in solution to yield fiber-like assemblies. This report begins with the discovery that 2EH-PDI can be trapped in a “dormant” conformation when hot solutions of the molecule in methylcyclohexane-chloroform are cooled. The molecules eventually nucleate to form fiber-like assemblies upon standing at room temperature.

This is a remarkable discovery. The authors use quantum mechanical calculations to map out the conformational states of the EH appendages. They identify a series of conformations where steric effects interfere with face-to-face association of the perylene units. They find conformational minima separated by barriers of 4 kT and one of 14 kT of energy. While I am not very knowledgeable about the calculations themselves, I can appreciate that barriers of this magnitude would retard conversion of 2EH-PDI to a conformation that would enable π -stacking and association. [Note that “conformation” is misspelled on line 180.]

The authors identify four different types of self-assembly that derive from the dormant 2EH-PDI:

- i) Upon standing, assembly to form long fiber-like structures occurs, as mentioned above. This presumably occurs via a homogeneous nucleation mechanism. By sonicating these fibers, the authors obtain “seed” fragment crystallites that they can use as nucleating agents.
- ii) When seed fragments are added to a solution of 2EH-PDI, supramolecular self-assembly begins immediately, with no lag period. The authors call this seeded growth process “homo seeding”, and the data in Figure 4 are used to characterize the growth process as one involving primary nucleation.
- iii) When the solution of dormant unimers is subjected to mild shear (for example, stirring), a new type of aggregate is formed. These are 3D aggregates, characterized by spiky protrusions emanating from what appears to be a dense core. The authors employ kinetic arguments to conclude that secondary nucleation plays a key role in the formation of these aggregates. The authors refer to these structures as spherulites, but it is not clear that the authors have a deep appreciation of what that means. I recommend that the authors read and cite a key review about spherulite formation from small molecules [Shtukenberg et al. *Spherulites*, *Chem. Rev.* 2012, 112, 1805–1838]. This review comments that “noncrystallographic branching lies at the

heart of the spherulite growth mechanism". This is something that the authors could examine in their system. It is interesting that the 3D structures in Figure 5 somewhat resemble an object seen in Figure 7f of the *Chem Reviews* article. From the distribution of sizes of the 3D objects presented in Figure 5f,g, the authors can conclude that nucleation leading to these structures occurs over time in competition with the growth process.

I would strongly urge the authors NOT to use the term "self-seeding" for shear-induced crystallization. This term has a well-defined meaning in polymer science. It has been used for more than half a decade and refers to a very different phenomenon specific to polymers. I cite two representative papers. 1) Reiter and coworkers: "Cloning polymer single crystals through self-seeding" *Nat Mater.* 2009 8(4):348-53. doi: 10.1038/nmat2405. 2) Tao et al., "Self-Seeding of Block Copolymers with a π -Conjugated Oligo(p-phenylenevinylene) Segment: A Versatile Route toward Monodisperse Fiber-like Nanostructures", *Macromolecules* 2018, 51, 2065–2075; doi.org/10.1021/acs.macromol.8b00046

- iv) When the solution of dormant 2EH-PDI unimers is treated with platelet assemblies formed by a different perylene diimide (PE-PDI), different types of aggregates form, characterized by deposition of secondary 2EH-PDI crystals on the platelet surface and protrusions from the edges. This process, via a spectroscopic/kinetic study, is also shown to involve secondary nucleation.

Overall, this is a very interesting study, well executed and well written. I recommend acceptance with minor revisions.

In addition to my suggestions above, there are other things that could be done to improve the presentation.

1. Many of the studies appear to involve monitoring changes in the absorbance at 515 nm. But the authors do not explicitly tell us what they do. The y-axis label on many plots is either $A@515$ nm or $\alpha@515$ nm. But these terms are never defined. A couple of sentences to define these terms and to explain how they are interpreted would be helpful.
2. Platelet formation through self-assembly of PE-PDI is not reported elsewhere in the literature. Some additional images to supplement those in Figure S14b would help. These could show the platelets prior to sonication. A few words in the caption about this self-assembly would be interesting.

Reviewer #2 (Remarks to the Author):

In this manuscript, the authors described the synthesis of various topological supramolecular polymer architectures utilizing living supramolecular polymerization (LSP) and secondary nucleation events of two perylene diimide (PDI) derivatives. One of the PDI derivatives formed dormant monomers in solution upon rapid cooling due to the existence of multiple conformations of side alkyl chains at high temperatures. These dormant monomers were further employed in homo-seed-induced LSP (homo-seeding), and in secondary nucleation-elongation triggered by mechanical stirring (self-seeding) or seed of the other PDI derivative (hetero-seeding). The LSP produced 1D supramolecular homopolymers with controllable length, and the secondary nucleation-elongation events afforded complex 3D- 3D-spherulites and scarf-like 2D-hetero-architecture, respectively. While LSP has been investigated extensively in many systems already, secondary nucleation in supramolecular polymerization is an emerging research topic. First, I would like to point out a couple of very important points as follows.

A) Lately, the distinction between supramolecular polymers and crystals has become less clear to me. Supramolecular polymers are perhaps polymers that elongate into meso-microscopic levels, much like polymer chains, or they could be dilute solutions that display non-Newtonian behaviors akin to polymer solutions, right? The intrigue lies in the formation of these regulated assemblies, which diverge from traditional crystals by undergoing nucleation and growth, or even secondary nucleation. This raises the question of whether we should indeed classify the authors system as a supramolecular polymer. It's common knowledge that crystal growth features a lag phase, more likely when using mixed solvents. The justification of considering the structures depicted in SEM images as supramolecular polymers is debatable. For instance, would AFM observation reveal the elongation of one-dimensional chains with a few nanometers wide into micrometer lengths, as with other suramolecular polymers? Can the authors find any macroscopic observation related to polymer solutions? If not, this research might gain more significance if substantially revised to focus on crystalline studies.

B) The assembly process of the aggregates is monitored through their absorption spectra. Upon examining the spectrum shown in Figure S2a, it is apparent that it considerably differs from the typical absorption change associated with perylene bisimide aggregates. Primarily, there's a noticeable reduction in overall absorbance. Moreover, monomer absorption persists, with only slight red-shifted absorption band by the aggregates. Based on my research experience on perylene bisimide dyes, such an unusual absorption spectrum suggests a heterogeneous solution. That is, microscale crystals, which do not affect absorption, could be forming, resulting in a solution-phase equilibrium between monomers and aggregates. It would be advisable for the authors to rigorously verify this point using microscopy and light scattering techniques. Even visual observations could prove to be significant. Should this be the case, a considerable reassessment of the absorption spectral data employed in scaling analyses and further interpretations would be necessary.

C) Assessing the XRD patterns of the aggregates under discussion would be advantageous. While XRD may not differentiate between supramolecular polymers and crystals, it could give valuable information about the system's crystallinity level.

D) Another major issue is that the obtained value (-1.40 ± 0.19) of the scaling exponent from the homo-

seeding growth kinetics is significantly more negative compared to those (γ should be at least around -0.5 (ideally $\gamma > -0.5$ in primary nucleation-elongation), as observed in JACS, 2021, 143, 11777–11787; JACS, 2023, 145, 22009–22018.) reported before for seed-induced elongation. Furthermore, the modelled growth kinetics using a seed-induced nucleation-elongation framework from amyloid software (<http://www.amylofit.ch.cam.ac.uk>) showed a poor fit well with the experimental data (Section S3, Figure S11, Table S1). The authors did not provide any valid justification for this discrepancy.

E) An important dearth is that the authors did not consider the shape change of the growth curves to distinguish primary nucleation-elongation and secondary nucleation-elongation events (JACS, 2021, 143, 11777–11787). This is an important point that should not be neglected for probing secondary nucleation in supramolecular polymerization. Furthermore, all the log-log plots (Fig. 5c and 7d, and Fig. 11a, SI) of half-time vs. monomer concentration are made from three data points. At least four data points should be used to make the plots to get better accuracy of γ values.

Overall although this topic should be of interest to the broad readership in the functional material field, the above points should be clarified in terms of scientific correctness. I would like to discuss the authors in the revised version to judge the credibility of the manuscript again and reach a final decision.

In addition, the following points should be addressed properly to improve the quality of the manuscript.

1. The solutions shown in the Fig. 2a seem to be unclear. This is also reflected in the high absorbance in most of the cases (including S1). The reliability of the spectroscopic results is of concern.
2. “The Uv-Vis absorption spectra of dormant monomers solution after adding the 10 mol% sonicated seed showed a sudden decrease in the monomeric absorption peaks at 515 nm and 495 nm, with an increase in the absorbance of the band at 575 nm (Figure 4b and S10).”

Comment: Fig. 4b shows that a sudden decrease in the absorption occurs after adding 5 mol% sonicated seed.

3. Text: “From FE-SEM studies, we found that the average length of the fibres prepared from [2EH-PDI monomer] / [2EH-PDI seed] ratios of 19:1, 3:1, and 1:1 are 6 μm , 2 μm , and 1 μm , respectively (Figure 4d-f).”

Figure caption: “(d), (e) and (f) after performing LSP to 2EH-PDI dormant monomers at varying [2EH-PDI monomer] / [2EH-PDI seed] ratios of 19:1, 3:1, and 1:1 respectively ($c = 50 \mu\text{M}$, $l = 10 \text{ mm}$, solvent = MCH*.)”

Comment: Fig. 4d-f are shown in opposite order. Correction should be done either in the figure or the text and caption.

4. “However, after stirring at 600 RPM, we could observe mostly 1D SPs. This could be due to the detachment of 1D SPs from spherulite structures, probably due to strong mechanical agitation at 600 RPM (Figure S12).”

Comment: It is necessary to measure γ at 600 rpm and compare with that at 60 rpm to know if secondary nucleation is affected by high-speed stirring. If remains unaffected, then it can be confirmed that the dispersed fibers are obtained due to detachment from spherulite structures.

5. Why does the size of the spherulite domains obtained at 120 rpm look larger than that at 60 rpm? Smaller domains are expected if fibers are detached upon increasing stirring speed.

6. Fig. 5c. log-log plot of the half-times of stir-induced (60 RPM) supramolecular polymerization versus the original concentration of 2EH-PDI.....

Comment: The concentration of seed should be mentioned.

7. Is there any difference in the stability of the hetero domains of the scarf-like 2D- architecture formed by hetero-seeded supramolecular polymerization? Is it possible to disassemble and reassemble one of them reversibly by external stimulus control keeping the other domain intact?

8. "Optical microscopy experiments were performed on an Olympus IX83 inverted fluorescence microscopy, and solution is taken in a transparent rectangular capillaries with 1 mm path length."

Comment: Since the dimension of the aggregates is large and PDI units should endow the aggregates with high fluorescence, the authors should also try visualization of secondary nucleation by fluorescence mode.

Other minor points:

1. "From FE-SEM studies, we found that the average length of the fibres prepared from [2EH-PDI monomer] / [2EH-PDI seed] ratios of 19:1, 3:1, and 1:1 are 6 μm , 2 μm and 1 μm , respectively (Figure 4d-f)."

Comment: Seed size should be mentioned in this discussion.

2. "We found the distant nucleation events during the supramolecular polymerization of dormant monomers through homo-seeding, self-seeding and hetero-seeding, resulting in the creation of elegant supramolecular architectures such as 3D spherulites and scarf-like SP heterostructures."

Comment: "distant nucleation events" should be changed to "distinct nucleation events"

3. "Fig. 11" in SI should be changed to "Fig. S11"

4. References should be checked properly. For example, volume is not bold in ref. 17. Similarly, in ref. 20, volume is italic instead of bold.

5. There are many formatting problems in the SI file. These should be checked properly.

6. d-spacings of the peaks should be assigned in Fig. S16.

Reviewer #3 (Remarks to the Author):

I co-reviewed this manuscript with one of the reviewers who provided the listed reports.

Reviewer #4 (Remarks to the Author):

In this work, authors reported the formation of 3D aggregates via living supramolecular polymerization. The design and experiments are helpful for the development of supramolecular chemistry. The reviewer suggested to accept this manuscript. However there are several important issues.

The main weak point of this manuscript lies in the absence of detailed information of macroscopic aggregates

1. Authors applied FE-SEM/TEM to confirm the existence of 3D structures. However, there is no structural information obtained from SEM/TEM. No connection between supramolecular polymerization and 3D aggregates was available.
2. The characterization of macroscopic aggregates was obviously insufficient. Authors did not provide any evidence of the chemical and physical properties of aggregates.

Point-to-Point responses to the reviewer's comments

We are grateful to all the reviewers for their efforts in going through our manuscript in detail and providing valuable comments. Their suggestions helped us to improve the quality of our manuscript further. We hope that the revised manuscript is suitable for publication in *Nature Communications*.

The changes made in the manuscript according to the reviewer's suggestions are marked in red colour.

Reviewer 1:

⇒ In this manuscript, the authors report studies of living supramolecular polymerization of two perylene diimide derivatives. The authors had previously shown that bis-2-ethylhexyl perylene diimide (2EH-PDI) undergoes cooperative supramolecular polymerization in solution to yield fiber-like assemblies. The molecules eventually nucleate to form fiber-like assemblies upon standing at room temperature.

This is a remarkable discovery. The authors use quantum mechanical calculations to map out the conformational states of the EH appendages. They identify a series of conformations where steric effects interfere with face-to-face association of the perylene units. They find conformational minima separated by barriers of 4 kT and one of 14 kT of energy. While I am not very knowledgeable about the calculations themselves, I can appreciate that barriers of this magnitude would retard conversion of 2EH-PDI to a conformation that would enable π -stacking and association. [Note that “conformation” is misspelled on line 180.]

Ans: We thank the reviewer for recognising the importance of our work and positive comments. We also thank the reviewer for finding the spelling mistake of `conformation`. This has now been corrected in the revised manuscript.

⇒ The authors identify four different types of self-assembly that derive from the dormant 2EH-PDI:

i) Upon standing, assembly to form long fiber-like structures occurs, as mentioned above. This presumably occurs via a homogeneous nucleation mechanism. By sonicating these fibers, the authors obtain “seed” fragment crystallites that they can use as nucleating agents.

ii) When seed fragments are added to a solution of 2EH-PDI, supramolecular self-assembly begins immediately, with no lag period. The authors call this seeded growth process “homo

seeding”, and the data in Figure 4 are used to characterize the growth process as one involving primary nucleation.

Ans: We thank the reviewer for nicely summarising our work on homo-seeding involving primary nucleation.

⇒ iii) When the solution of dormant unimers is subjected to mild shear (for example, stirring), a new type of aggregate is formed. These are 3D aggregates, characterized by spiky protrusions emanating from what appears to be a dense core. The authors employ kinetic arguments to conclude that secondary nucleation plays a key role in the formation of these aggregates. The authors refer to these structures as spherulites, but it is not clear that the authors have a deep appreciation of what that means. I recommend that the authors read and cite a key review about spherulite formation from small molecules [Shtukenberg et al. Spherulites, *Chem. Rev.* 2012, 112, 1805–1838]. This review comments that “noncrystallographic branching lies at the heart of the spherulite growth mechanism”. This is something that the authors could examine in their system. It is interesting that the 3D structures in Figure 5 somewhat resemble an object seen in Figure 7f of the *Chem Reviews* article. From the distribution of sizes of the 3D objects presented in Figure 5f,g, the authors can conclude that nucleation leading to these structures occurs over time in competition with the growth process.

Ans: We thank the reviewer for these valuable comments on Spherulites. In the present case, the supramolecular polymers of **2EH-PDI** do not possess good crystalline order (See Figure S23a). "X" shape morphology and low-angle branching are commonly observed at the initial stages of spherulite formation. When we probed the temporal evolution of **2EH-PDI** by applying mechanical stimuli such as repetitive pipetting, we observed the formation of "X" shape morphology (hedrite) and small angle branching. These observations further support that the observed morphology is a spherulite. Based on reviewer suggestions, we have further expanded the discussion on spherulite structures and cited the new references as references 53 and 61.

The following sentences were added/modified in the revised manuscript in page number 11, bottom paragraph. The new references were added as Ref. 50 and 61 in the revised manuscript.

"The formation of intermediate “X” shape (hedrite) morphology in the initial stages of growth is a characteristic of spherulite formation.^{50,61} With the increase in time, we have observed fibril growth in all directions via small-angle branching,^{50,61} resulting in the formation of star-shaped morphology (Figure 6e,f)."

50. Shtukenberg, A. G., Punin, Y. O., Gunn, E. & Kahr, B. Spherulites. *Chem. Rev.* **112**, 1805–1838 (2012).
61. Xiao, Y. *et al.* Growth mechanism of the spherulitic propylthiouracil-kaempferol cocrystal: new perspectives into surface nucleation. *CrystEngComm* **23**, 2367–2375 (2021).

⇒ I would strongly urge the authors NOT to use the term “self-seeding” for shear-induced crystallization. This term has a well-defined meaning in polymer science. It has been used for more than half a decade and refers to a very different phenomenon specific to polymers. I cite two representative papers. 1) Reiter and coworkers: “Cloning polymer single crystals through self-seeding” *Nat Mater.* 2009 8(4):348-53. doi: 10.1038/nmat2405. 2) Tao et al., “Self-Seeding of Block Copolymers with a π -Conjugated Oligo(p-phenylenevinylene) Segment: A Versatile Route toward Monodisperse Fiber-like Nanostructures”, *Macromolecules* 2018, 51, 2065–2075; doi.org/10.1021/acs.macromol.8b00046

Ans: We are thankful to the reviewer for this comment. We agree that self-seeding is not an appropriate word to describe the supramolecular polymerization of **2EH-PDI** via shear forces. In the revised manuscript, we have removed the word "self-seeding" and instead used "shear-induced assembly or shear forces".

⇒ iv) When the solution of dormant 2EH-PDI unimers is treated with platelet assemblies formed by a different perylene diimide (PE-PDI), different types of aggregates form, characterized by deposition of secondary 2EH-PDI crystals on the platelet surface and protrusions from the edges. This process, via a spectroscopic/kinetic study, is also shown to involve secondary nucleation.

Overall, this is a very interesting study, well executed and well written. I recommend acceptance with minor revisions.

Ans: We are grateful to the reviewer for recognising the importance of our work, providing constructive comments and recommending it for publication.

⇒ In addition to my suggestions above, there are other things that could be done to improve the presentation.

1. Many of the studies appear to involve monitoring changes in the absorbance at 515 nm. But the authors do not explicitly tell us what they do. The y-axis label on many plots is either

$A@515$ nm or $\alpha@515$ nm. But these terms are never defined. A couple of sentences to define these terms and to explain how they are interpreted would be helpful.

Ans: We thank the reviewer for bringing this point. Accordingly, we have defined and explained both $A@515$ nm and $\alpha@515$ nm in the revised manuscript.

The following sentences were added/modified in the revised manuscript. See page 4, results section

"We have studied the supramolecular polymerization and depolymerization process of **2EH-PDI** by monitoring its degree of polymerization (α) at 515 nm, ($\alpha @ 515$ nm). Above the elongation temperatures (T_e), α is zero as **2EH-PDI** (50 μ m) exists as monomers (Figure S2b)."

"Probing the absorbance (A) at 515 nm ($A @ 515$ nm) indicates that by increasing the percentage of good solvent, such as DCE, the temperature region of the hysteresis loop enhanced (Figure 2b and Figures S3-S6). Monomers of **2EH-PDI** have a higher absorbance at 515 nm than its SPs. Hence, the decrease in the absorbance at 515 nm is an indication of supramolecular polymerization of **2EH-PDI** (Figure 2b and Figures S3-S6).⁴⁹"

⇒ 2. Platelet formation through self-assembly of PE-PDI is not reported elsewhere in the literature. Some additional images to supplement those in Figure S14b would help. These could show the platelets prior to sonication. A few words in the caption about this self-assembly would be interesting.

Ans: We thank the reviewer for this comment. In the revised manuscript we have included the FE-SEM images of **PE-PDI** before and after sonication in Figure S21 and described its self-assembly in the figure caption.

The following things were added in the revised manuscript.

Figure S21. (a) Electronic UV-vis absorption spectra of **PE-PDI** in DCE (red) and MCH* (blue). FE-SEM images of **PE-PDI** (b) before and (c) after sonication for 1 hour ($c = 25 \mu\text{M}$, $l = 10 \text{ mm}$ Solvent: MCH* at 303K).

"The UV-vis spectra suggest that **PE-PDI** (25 μM) in DCE exhibits a well-resolved absorption spectrum with vibrational features, indicating its monomeric form at 30 °C. However, in MCH*, the spectra show lower absorbance, and the emergence of a new red-shifted band centered at 575 nm, suggests the self-assembly of **PE-PDI** (Figure S21a). FE-SEM studies of **PE-PDI** in MCH* reveal that the self-assembled aggregates of **PE-PDI** have 2D platelets like morphology (Figure S21b)."

Reviewer 2:

⇒ In this manuscript, the authors described the synthesis of various topological supramolecular polymer architectures utilizing living supramolecular polymerization (LSP) and secondary nucleation events of two perylene diimide (PDI) derivatives. One of the PDI derivatives formed dormant monomers in solution upon rapid cooling due to the existence of multiple conformations of side alkyl chains at high temperatures. These dormant monomers were further employed in homo-seed-induced LSP (homo-seeding), and in secondary nucleation-elongation triggered by mechanical stirring (self-seeding) or seed of the other PDI derivative (hetero-seeding). The LSP produced 1D supramolecular homopolymers with controllable length, and the secondary nucleation-elongation events afforded complex 3D- 3D-spherulites and scarf-like 2D-hetero-architecture, respectively. While LSP has been investigated extensively in many systems already, secondary nucleation in supramolecular polymerization is an emerging research topic. First, I would like to point out a couple of very important points as follows.

Ans: We are very thankful to the reviewer for recognising the importance of secondary nucleation in supramolecular polymerizations and providing valuable insights to improve the quality of our manuscript further.

⇒ A) Lately, the distinction between supramolecular polymers and crystals has become less clear to me. Supramolecular polymers are perhaps polymers that elongate into meso-microscopic levels, much like polymer chains, or they could be dilute solutions that display non-Newtonian behaviors akin to polymer solutions, right? The intrigue lies in the formation of these regulated assemblies, which diverge from traditional crystals by undergoing nucleation and growth, or even secondary nucleation. This raises the question of whether we should indeed classify the authors system as a supramolecular polymer. It's common knowledge that crystal growth features a lag phase, more likely when using mixed solvents. The justification of considering the structures depicted in SEM images as supramolecular polymers is debatable. For instance, would AFM observation reveal the elongation of one-

dimensional chains with a few nanometers wide into micrometer lengths, as with other supramolecular polymers? Can the authors find any macroscopic observation related to polymer solutions? If not, this research might gain more significance if substantially revised to focus on crystalline studies.

Ans: We thank the reviewer for these comments. Small molecules that assemble into one-dimensional (1D) structures via various directional non-covalent interactions in dilute or semi-dilute solutions are called supramolecular polymers (SPs) (See the latest review: By Aida and Meijer, *Isr. J. Chem.* **60**, 33-47 (2020)). Several reports suggest that the supramolecular polymers assemble into 1D chains with sizes much larger than the mesoscopic scale. For example, supramolecular polymers of phthalocyanine based hydrogen-bonded π -systems have a width and length of several micrometres (See *Nature Materials* **21**, 253–261 (2022)). Organometallic complexes are reported to form supramolecular polymers with rigid fibres-like morphology having widths and lengths of several microns (See *Nat Commun* **14**, 1084 (2023)). The macroscopic single crystals composed of 1D chains are also considered as supramolecular polymers (See *J. Am. Chem. Soc.* **142**, 16557–16561 (2020,); *Angew. Chemie - Int. ed.* **50**, 1397-1401 (2011)). Usually, supramolecular polymers having hydrogen-bonding interactions are known to form one-dimensional chains with a few nanometers wide and micrometer lengths. However, as described above, the dimensions of other types of supramolecular polymers formed by monomers having no hydrogen-bonding groups can be much larger. Please note that the supramolecular polymerization of **2EH-PDI** is well characterized by us using various spectroscopic, microscopic and theoretical studies in our previous paper, *Chem. - An Asian J.* **17**, e202200494 (2022), which is Ref. 49 in the current manuscript. In short, **2EH-PDI** forms 1D assemblies driven by directional non-covalent interactions such as π - π stacking in dilute solutions. These 1D assemblies do not possess good crystalline order and are flexible. FE-SEM, TEM and AFM studies show that the width of these SPs is < 500 nm, and the length is several micrometres (See *Chem. - An Asian J.* **17**, e202200494 (2022) and Figures S8&S20d in the current manuscript). We have also shown that the mechanism of formation of SPs by **2EH-PDI** fits well with the models developed for SPs (*Chem. - An Asian J.* **17**, e202200494 (2022)). Hence, crystallization may not be appropriate to describe the 1D self-assembly of **2EH-PDI**. Based on the definition of supramolecular polymers and several literature reports, we can confidently say that the 1D structures of **2EH-PDI** described in this manuscript are supramolecular polymers.

⇒ B) The assembly process of the aggregates is monitored through their absorption spectra. Upon examining the spectrum shown in Figure S2a, it is apparent that it considerably differs from the typical absorption change associated with perylene bisimide aggregates. Primarily, there's a noticeable reduction in overall absorbance. Moreover, monomer absorption persists, with only slight red-shifted absorption band by the aggregates. Based on my research experience on perylene bisimide dyes, such an unusual absorption spectrum suggests a heterogeneous solution. That is, microscale crystals, which do not affect absorption, could be forming, resulting in a solution-phase equilibrium between monomers and aggregates. It would be advisable for the authors to rigorously verify this point using microscopy and light scattering techniques. Even visual observations could prove to be significant. Should this be the case, a considerable reassessment of the absorption spectral data employed in scaling analyses and further interpretations would be necessary.

Ans: We thank the reviewer for this comment. For various PDI molecules, considerable reduction in the overall absorbance is observed upon self-assembly (Please see: *Org. Lett.* **9**, 1085–1088 (2007); *J. Am. Chem. Soc.* **137**, 3924–3932 (2015)). We have seen similar behaviour in alkyl PDIs (**2EH-PDI**) as well (*Chem. - An Asian J.* **17**, e202200494 (2022)). We found that the solutions of supramolecular polymers of **2EH-PDI** prepared via homo-seeding do not show any visible inhomogeneities (See Figure S16a). However, after the formation of spherulites by the mechanical agitation the solutions show inhomogeneities (See Figure S16a). This is because of the much larger-sized 3D structures of spherulites compared to 1D fibers, as evidenced by FE-SEM images (Figures 5e and S14). Hence, we have used the absorbance at the monomeric wavelength (515 nm or 480 nm) to probe the mechanism and scaling analysis, because monomers always stay in solution. A decrease in monomeric absorbance is also a sign of supramolecular polymerization, which we and others have previously used to probe the supramolecular polymerization mechanisms of perylene diimides (See ref *J. Am. Chem. Soc.* **137**, 3300–3307 (2015); *J. Am. Chem. Soc.* **137**, 3924–3932 (2015); *Chem. - An Asian J.* **17**, e202200494 (2022)). Since we have used a decrease in monomeric absorbance to probe the mechanisms and scaling analyses, the problems caused by heterogeneities in the solutions (for spherulites) can be avoided.

Part of the Figure S16a is shown below

Figure S16: (a) Photographs of solutions containing 1D fibres (left) and 3D spherulites (right) of **2EH-PDI**.

⇒ C) Assessing the XRD patterns of the aggregates under discussion would be advantageous. While XRD may not differentiate between supramolecular polymers and crystals, it could give valuable information about the system's crystallinity level.

Ans: We appreciate the reviewer for this comment. The SPs of **2EH-PDI** showed a broad and low intense peak at $2\theta = 4.31^\circ$, corresponding to a d-spacing of 2.05 nm, indicating a poor crystalline order (Figure S23). On the other hand, the 2D platelets of **PE-PDI** showed good crystalline order, as evidenced by several sharp and high-intensity peaks. The first peak was observed at $2\theta = 5.54^\circ$, corresponding to a d-spacing of 1.6 nm. Successive peaks at 0.8 nm ($2\theta = 11.09^\circ$) (d/2), 0.54 nm ($2\theta = 16.45^\circ$) (d/3), and 0.44 nm ($2\theta = 19.96^\circ$) ($\sim d/4$) are also observed (Figure S23). This indicates the lamellar-like packing of **PE-PDI** molecules with interdigitation of side chains in the 2D platelets. Identifying the minor peak at 0.7 nm ($2\theta = 12.48^\circ$) remains inconclusive, which can be estimated by determining the unit cell, which is beyond the scope of the present study.

The following sentences and Figure S23 are included in the revised manuscript. See page 14, bottom paragraph of the manuscript.

"The SPs of **2EH-PDI** showed a broad and low intense peak at $2\theta = 4.31^\circ$ (d-spacing of 2.05 nm) indicating a poor crystalline order. On the other hand, the 2D platelets of **PE-PDI** showed good crystalline order, as evidenced by several sharp and high-intensity peaks. The first peak was observed at $2\theta = 5.54^\circ$, corresponding to a d-spacing of 1.6 nm. Successive peaks at 0.8 nm ($2\theta = 11.09^\circ$) (d/2), 0.54 nm ($2\theta = 16.45^\circ$) (d/3), and 0.44 nm ($2\theta = 19.96^\circ$) ($\sim d/4$) are also observed. This indicates the lamellar-like packing of **PE-PDI** molecules with interdigitation of side chains in the 2D platelets (Figure S23)."

Figure S23. (a) Thin film XRD pattern of **2EH-PDI** (red), **PE-PDI** (blue) and SP heterostructures (green) obtained via hetero seeding approach by adding 50 mol% **PE-PDI** seeds to **2EH-PDI** dormant monomers (50 μM) in MCH*. (b) Schematic illustration of proposed molecular packing of **PE-PDI** molecules in the 2D platelets.

⇒D) Another major issue is that the obtained value (-1.40 ± 0.19) of the scaling exponent from the homo-seeding growth kinetics is significantly more negative compared to those (γ should be at least around -0.5 (ideally $\gamma > -0.5$ in primary nucleation-elongation), as observed in JACS, 2021, 143, 11777–11787; JACS, 2023, 145, 22009–22018.) reported before for seed-induced elongation. Furthermore, the modelled growth kinetics using a seed-induced nucleation-elongation framework from amyloid software (<http://www.amylofit.ch.cam.ac.uk>) showed a poor fit well with the experimental data (Section S3, Figure S11, Table S1). The authors did not provide any valid justification for this discrepancy.

Ans: We thank the reviewer for bringing this comment. According to the previous reports, a strong monomer-dependent primary or secondary nucleation process is signified by $|\gamma| \geq 1$, unless it pertains to a fragmentation process, in which case a γ value close to 0.5 is observed (See references 57 and refer to the second paragraph on page 9759 of reference 51). In the present case, we have not observed any fragmentation and it is a monomer-dependent process where $|\gamma|$ can be greater than 1. In the case of homo-seeding, our analysis yielded $|\gamma| = 1.2 \pm 0.012$, derived from a fitting procedure involving four data points. Subsequently, our investigation revealed that the seed-induced primary nucleation-elongation framework provided the best fit, as illustrated in Figure S11 and the accompanying Table S1. This data showed a less favourable fit for the seed-induced secondary nucleation-elongation model, as

presented in Figure S12 and the corresponding Table S1. Furthermore, upon introducing the seed, we observed a remarkable spontaneous growth in kinetics, characterized by the absence of a lag phase and manifestation of non-sigmoidal transition (See Figure 5b). These findings collectively support the conclusion that a homo-seeding process occurs through the seed-induced nucleation-elongation mechanism.

Accordingly, the following paragraph has been modified, and a new reference (Ref. 57) was added to the revised manuscript.

"To obtain a clear understanding of the molecular mechanism, we conducted kinetic analyses at various concentrations of dormant monomer, specifically 40 μM , 45 μM , 50 μM and 55 μM while keeping seed concentration constant at 12.5 μM . We monitored the growth kinetics by measuring changes in the absorbance at 480 nm ($A @ 480 \text{ nm}$). The scaling exponent, which characterizes how the reaction's lag time or half-time (t_{50}) varies with the initial monomer concentration, was determined by the double-logarithmic plot of t_{50} against monomer concentration, as depicted in Figure S11a under seeded conditions.²⁴⁻³⁰ The obtained value for the scaling exponent is $\gamma = -1.20 \pm 0.09$, corresponding to reaction order $n_1 = 2.4$. In the present case, we have not observed any fragmentation, and it is a monomer-dependent process hence γ can be a more negative value than -1.^{51,57} The growth kinetics were modelled using a seed-induced nucleation-elongation framework from amyloid software (<http://www.amylofit.ch.cam.ac.uk>)³⁰ and fit well with the experimental data (Section S3, Figure S11, Table S1). However, the same kinetic data showed a less favourable fit for the secondary nucleation-elongation model (Figure S12, Table S1). The above observations clearly indicate that the conversion of dormant monomers of 2EH-PDI into SPs via homo-seeding takes place through the primary nucleation-elongation mechanism."

⇒ E) An important dearth is that the authors did not consider the shape change of the growth curves to distinguish primary nucleation-elongation and secondary nucleation-elongation events (JACS, 2021, 143, 11777–11787). This is an important point that should not be neglected for probing secondary nucleation in supramolecular polymerization. Furthermore, all the log-log plots (Fig. 5c and 7d, and Fig. 11a, SI) of half-time vs. monomer concentration are made from three data points. At least four data points should be used to make the plots to get better accuracy of γ values.

Ans: We thank the reviewer for bringing up this point. In the revision, we have comprehensively compared the growth kinetics between homo-seeding and 60 RPM (Figure 5b). This analysis reveals that the growth kinetics at 60 RPM exhibited a sigmoidal nature, accompanied by the presence of a discernible lag time. In stark contrast, when considering homo-seeding, we observed spontaneous growth in kinetics, characterized by a non-sigmoidal nature and the absence of any lag phase (Figure 5b). We have also meticulously recalculated

the γ values for all experiments using a dataset comprising four data points (Figures S11a, 5c, and 7d).

The following sentence is included in the revised manuscript, see page 9, bottom paragraph. Part of Figure 5, which is included in the manuscript, is also shown below.

"In stark contrast, when considering homo-seeding, we observed spontaneous growth in kinetics, characterized by a non-sigmoidal nature and the absence of any lag phase (Figure 5b)."

Fig. 5: Formation of 3D spherulites via stirring-induced secondary nucleation. **b** Illustration of differences between primary nucleation (homo-seeding) and secondary nucleation (stirring at 60 RPM) events. The sigmoidal growth of the **2EH-PDI** dormant monomer under stirring indicates the presence of the secondary nucleation process. **c** log-log plot of the half-times of stir-induced (60 RPM) supramolecular polymerization versus the original concentration of **2EH-PDI**. Symbols represent the experimental data; solid line is a power law fit. This fit shows a linear trend with a slope of -3.00 ± 0.37 referred to as the exponent coefficient (γ), indicating a monomer dependent secondary nucleated supramolecular polymerization process. **Since there is no seed is added externally, the concentration of the seed is zero.** **d** Kinetic profiles of the concentration-dependent experiments at 60 RPM were obtained by monitoring the absorbance at 480 nm while varying the dormant monomer (**2EH-PDI**) concentrations to 40 μM , 45 μM , 50 μM and 55 μM .

⇒ Overall although this topic should be of interest to the broad readership in the functional material field, the above points should be clarified in terms of scientific correctness.

I would like to discuss the authors in the revised version to judge the credibility of the manuscript again and reach a final decision. In addition, the following points should be addressed properly to improve the quality of the manuscript.

Ans. We thank the reviewer for recognising the importance of our work and providing valuable suggestions. We hope that the reviewer will be convinced by the revised version of the manuscript.

⇒1. The solutions shown in the Fig. 2a seem to be unclear. This is also reflected in the high absorbance in most of the cases (including SI). The reliability of the spectroscopic results is of concern.

Ans. We thank the reviewer for this comment. In the revised version of the manuscript, we replaced the solution photos of **2EH-PDI** with higher-resolution images (Figure 2a). The instrument model utilized for the absorption spectra was the Jasco V-770 spectrophotometer which can measure absorbance up to 4 in the Uv-Visible region. All our samples display absorbance well below this value. Since this study involves various seeding conditions and mechanical agitation (stirring and pipetting), we have performed all the experiments in 1 cm path length (3 mL) quartz cuvettes. As a result, the absorbance values at 515 nm are higher. For concentration-dependent experiments, we monitored at 480 nm, where the absorbance is further lower than 515 nm, and both the kinetic profiles look identical (See the figure below). Please note that in the previous manuscript version, for concentration-dependent experiments, we mistakenly mentioned that it was monitored at 515 nm (A @ 515 nm) instead of 480 nm (A @ 480 nm). This has been corrected in the revised manuscript (See Figures 5b,d, 7e, S11, S12, S15, S18, S19).

The modified Figure 2a is shown below.

Fig. 2: Synthesis of dormant monomer. **a** Synthesis of dormant monomers of **2EH-PDI**. Heating the supramolecular polymers (SPs) formed at 303 K to 353 K leads to the formation of monomers in MCH*. Later by rapid cooling (10 K/min.) to 303 K it led to the formation of dormant monomers. When these dormant monomers were left undisturbed at 303 K, again, SPs were formed after 90 minutes.

The following figure shows the comparison between kinetic data recorded by monitoring absorbance at 515 nm and 480 nm.

⇒2. The Uv-Vis absorption spectra of dormant monomers solution after adding the 10 mol% sonicated seed showed a sudden decrease in the monomeric absorption peaks at 515 nm and 495 nm, with an increase in the absorbance of the band at 575 nm (Figure 4b and S10).”

Comment: Fig. 4b shows that a sudden decrease in the absorption occurs after adding 5 mol% sonicated seed.

Ans. We thank the reviewer for pointing out this mistake. In the revised manuscript, the data corresponding to 5 mol% of the seed is now presented as Figure S10.

Figure S10. Time dependent absorption spectral changes of 2EH-PDI after addition of 5 mol% prefabricated 2EH-PDI aggregates ($c = 50 \mu\text{M}$) after sonicating 1 hour at 303 K temperature. ($c = 50 \mu\text{M}$, $l = 10 \text{ mm}$, Solvent: MCH*).

⇒3. Tex. “From FE-SEM studies, we found that the average length of the fibres prepared from [2EH-PDI monomer] / [2EH-PDI seed] ratios of 19:1, 3:1, and 1:1 are 6 μm, 2 μm, and 1 μm, respectively (Figure 4d-f).” Figure caption: “(d), (e) and (f) after performing LSP to 2EH-PDI dormant monomers at varying [2EH-PDI monomer] / [2EH-PDI seed] ratios of 19:1, 3:1, and 1:1 respectively ($c = 50 \mu\text{M}$, $l = 10 \text{ mm}$, solvent = MCH*.)”

Comment: Fig. 4d-f are shown in opposite order. Correction should be done either in the figure or the text and caption.

Ans. We thank the reviewer for pointing out this mistake. We have corrected the Figure caption in the revised manuscript.

⇒4. “However, after stirring at 600 RPM, we could observe mostly 1D SPs. This could be due to the detachment of 1D SPs from spherulite structures, probably due to strong mechanical agitation at 600 RPM (Figure S12).”

Comment: It is necessary to measure γ at 600 rpm and compare with that at 60 rpm to know if secondary nucleation is affected by high-speed stirring. If remains unaffected, then it can be confirmed that the dispersed fibers are obtained due to detachment from spherulite structures.

Ans. We thank the reviewer for bringing up this point. Initially, we thought that the formation of 1D fibres at 600 rpm could be due to the detachment of fibres from spherulite structures. However, after the reviewer comments, we have thoroughly investigated this point once again. First, we prepared the spherulite structures by stirring at 60 RPM, and then they were subjected to stirring at 600 RPM for 1 hour. Subsequent FE-SEM studies confirmed the stability of these spherulites even at the higher stirring speed of 600 RPM (Figure S17). Next, we conducted kinetic growth studies of **2EH-PDI** at 600 RPM across four different concentrations and obtained a scaling exponent (γ) close to -2.0 (Figure S18a). The primary nucleation reaction order (n_c) and secondary nucleation reaction order (n_2) were determined to be 4 and 3, respectively. Notably, the kinetic data showed a good fit for both the unseeded primary nucleation-elongation model (Figure S18) and the secondary nucleation-elongation model (Figure S19). This suggests that supramolecular polymerization at 600 RPM may be driven by both primary and secondary nucleation events resulting in the formation of 1D fibers and 3D spherical spherulites. This phenomenon is also observed in protein aggregation, particularly in

amyloid formation, where both primary and secondary nucleation events take place (See Ref. 54, 59 and 60).

Following paragraph and Figures are added to the revised manuscript. See page 11, second paragraph in the revised manuscript.

"We noticed that up to 300 RPM, we could observe the formation of 3D spherical spherulites, and the size of spherulites decreased with increasing the RPM (Figures 5e,f and S14a-c). However, after stirring at 600 RPM, we could observe mostly 1D SPs and a few spherulites (Figure 5g and S14d). To understand the stability of spherulites at 600 RPM, we synthesised them at 60 RPM and stirred them at 600 RPM for 1 hour. Interestingly, FE-SEM data reveal that these spherulites are stable even when the solution is subjected to strong mechanical agitation up to 600 RPM (Figure S17). However, when the dormant monomers of **2EH-PDI** were stirred at 600 RPM, mostly 1D SPs were formed. To understand this further, we conducted kinetic growth studies of **2EH-PDI** at 600 RPM across four different concentrations and obtained a scaling exponent (γ) close to -2.0 (Figure S18a). The primary nucleation reaction order (n_c) and secondary nucleation reaction order (n_2) were determined to be 4 and 3, respectively. Notably, the kinetic data showed a good fit for both the unseeded primary nucleation-elongation model (Figure S18) and the secondary nucleation-elongation model (Figure S19). This suggests that supramolecular polymerization at 600 RPM is driven by both primary and secondary nucleation events, resulting in the formation mixture of 1D fibers and 3D spherical spherulites (Figure S14d). This phenomenon was also observed in protein aggregation, particularly in amyloid formation, where both primary and secondary nucleation events occur.^{51,59,60}"

Figure S18. a) (c) log-log plot of the half-times of stir-induced (600 RPM) supramolecular polymerization versus the original concentration of **2EH-PDI**. Symbols represent the experimental data; solid line is a power law fit. This fit shows a linear trend with a slope of -2.00 ± 0.374 referred to as the exponent coefficient (γ), indicating a monomer dependent supramolecular polymerization process. (b-e) Fitting of shear-induced kinetics at 600 RPM conditions into unseeded primary nucleation-elongation model by using online software <http://www.amylofit.ch.cam.ac.uk> at various

dormant monomer concentration: b) 40 μM , c) 45 μM , d) 50 μM and e) 60 μM ($\alpha @ 480 =$ degree of supramolecular polymerization monitored at 480 nm, solvent: MCH*, $l = 10$ mm at 303 K).

Figure S19. a-d) Fitting of shear-induced kinetics at 600 RPM conditions into unseeded secondary nucleation-elongation model by using online software <http://www.amylofit.ch.cam.ac.uk> at various dormant monomer concentrations of **2EH-PDI**: a) 40 μM , b) 45 μM , c) 50 μM and d) 60 μM ($\alpha @ 480 =$ degree of supramolecular polymerization monitored at 480 nm, solvent: MCH*, $c = 50$ μM , $l = 10$ mm at 303 K).

59. Haass, C. & Selkoe, D. J. Soluble protein oligomers in neurodegeneration: Lessons from the Alzheimer's amyloid β -peptide. *Nat. Rev. Mol. Cell Biol.* **8**, 101–112 (2007).
60. Zimmermann, M. R. et al. Mechanism of secondary nucleation at the single fibril level from direct observations of A β 42 aggregation. *J. Am. Chem. Soc.* **143**, 16621–16629 (2021).

⇒5. Why does the size of the spherulite domains obtained at 120 rpm look larger than that at 60 rpm? Smaller domains are expected if fibers are detached upon increasing stirring speed.

Ans. We thank the reviewer for this comment. Due to the different scale bars, it looks like the spherulite domains formed at 120 RPM have a larger size than at 60 RPM. In the revised version, we have provided the uniform scale bar (Figure 5e,f) and analysed the size of several

spherulites using FE-SEM images (Figure S14). This suggests that the diameters of spherulites obtained at 60 RPM are larger, followed by those formed at 120 RPM and 300 RPM.

Following Figure is added to the supporting information of the revised manuscript.

Figure S14. FE-SEM images of **2EH-PDI** obtained after stirring the dormant monomers with various RPM (a) 60 RPM, (b) 120 RPM, (c) 300 RPM and (d) 600 RPM in MCH*. Inset: FE-SEM based histogram of the contour diameter of randomly selected 10 spherulites formed via 60 RPM (a), 120 RPM (b) and 300 RPM (c). The white circles in (d) represent the spherulite structures. ($c = 50 \mu\text{M}$, solvent: MCH* at 303K).

⇒6. Fig. 5c. log-log plot of the half-times of stir-induced (60 RPM) supramolecular polymerization versus the original concentration of 2EH-PDI.....

Comment: The concentration of seed should be mentioned.

Ans. We thank the reviewer for this comment. All measurements conducted under stirring conditions are unseeded experiments where the seed concentration is zero. This has been now mentioned in the figure caption of 5c of the revised manuscript.

⇒7. Is there any difference in the stability of the hetero domains of the scarf-like 2D- architecture formed by hetero-seeded supramolecular polymerization? Is it possible to disassemble and reassemble one of them reversibly by external stimulus control keeping the other domain intact?

Ans. We thank the reviewer for this suggestion. Accordingly, we have conducted additional experiments to disassemble one of the components of scarf-like hetero-architectures selectively. Interestingly, we found that SPs of **2EH-PDI** have less thermal stability than **PE-PDI** 2D platelets in 10% DCE in MCH (MCH*) solution. The SPs **2EH-PDI** showed melting at 307 K and completely depolymerized in the temperature range of 335 K to 340 K (Figure S26a,b). In contrast, **PE-PDI** seeds are stable up to 323 K (Figure S26a,c). Taking advantage of differences in thermal stability, the SP heterostructure solution was heated at 323 K for 15 minutes, and this hot solution was coated on the silicon wafer. FE-SEM images revealed the absence of SP heterostructure and the presence of segregated SPs of **2EH-PDI** and 2D platelets of **PE-PDI** (Figure S26d). This indicates that at 323 K, SPs of **2EH-PDI** are melted and detached from the surface of 2D platelets of **PE-PDI** and formed separately, probably during solvent evaporation. On the other hand, when this solution was cooled from 323 K to 303 K, and coated on silicon wafer after waiting for an hour, we observed the formation of SP heterostructures (Figure S26e,f). Furthermore, we immersed the silicon wafer containing SP heterostructures in MCH* solvent at 323 K for various time intervals and visualized them through FE-SEM, which showed only 2D platelet morphology (Figure S27). These observations indicate that SPs of **2EH-PDI** can be selectively removed and reattached to the surface of **PE-PDI** seeds.

The following paragraph and Figures are added to the revised manuscript. See page 15, bottom paragraph of the revised manuscript.

"Next, we have also explored the possibility of selectively disassembling one of the components from scarf-like 2D heterostructure. Interestingly, we found that SPs of **2EH-PDI** have less thermal stability than **PE-PDI** 2D platelets in MCH*. The SPs **2EH-PDI** showed melting at 307 K and significant depolymerization was observed at 323 K (Figure S26a,b). In contrast, **PE-PDI** seed particles are stable up to 330 K in MCH* (Figure S26a,c). Taking advantage of differences in thermal stability, the SP heterostructure solution was heated at 323 K for 15 minutes, and this hot solution was coated on the silicon wafer. FE-SEM images revealed the absence of SP heterostructure and the presence of segregated SPs of **2EH-PDI** and 2D platelets of **PE-PDI** (Figure S26d). This indicates that at 323 K, SPs of **2EH-PDI** are melted and detached from the surface of 2D platelets of **PE-PDI** and formed separately, probably during solvent evaporation. On the other hand, when this solution was cooled from 323 K to 303 K, and

coated on a silicon wafer after waiting for an hour at 303 K, we observed the formation of SP heterostructures from FE-SEM images (Figure S26e,f). Furthermore, we immersed the silicon wafer containing SP heterostructures in MCH* solvent at 323 K for various time intervals and visualized them through FE-SEM, which showed only 2D platelet morphology (Figure S27b-d). This is due to the selective depolymerization and dissolution of SPs of **2EH-PDI** at 323 K from the surface of **PE-PDI** seeds into MCH* solution."

Figure S26. (a) Melting analysis of **2EH-PDI** 1D fibers (50 μM , MCH*) and **PE-PDI** seed particles prepared by sonicating for 1 hour (25 μM , MCH*). (b) UV-vis spectra of **2EH-PDI** 1D fibres (50 μM , MCH*) at 303 K (blue) and 323 K (red). (c) UV-vis spectra of **PE-PDI** seed particles (25 μM , MCH*) at 303 K (blue) and 323 (red). (d) The FE-SEM image obtained by spin coating the solution containing SP heterostructures, which was heated at 323 K for 15 minutes. (e) and (f) FE-SEM images showing regeneration of SP heterostructures formed after cooling down the solution from 323 K to 303 and waiting for 1 hour.

Figure S27. (a) FE-SEM image of obtained SP heterostructures via hetero seeding approach (50 mol% of PE-PDI seed was added to the dormant monomers of 2EH-PDI). FE-SEM images (b), (c) and (d) depict the removal of 2EH-PDI fibers from SP heterostructures after dipping in MCH* at 323 K for various time intervals.

⇒8. Optical microscopy experiments were performed on an Olympus IX83 inverted fluorescence microscopy, and solution is taken in a transparent rectangular capillaries with 1 mm path length.”

Comment: Since the dimension of the aggregates is large and PDI units should endow the aggregates with high fluorescence, the authors should also try visualization of secondary nucleation by fluorescence mode.

Ans. We thank the reviewer for this comment. Accordingly, we have also visualized the secondary nucleation via hetero-seeding by fluorescence mode. In the revised manuscript, we have added the fluorescence microscopy images as Figure S25 and the video (Movie: Hetero-seeding-fluorescence) is also provided.

Figure S25. Fluorescence microscopy images showing time-dependent growth of SP heterostructures after mixing the 50 mol% of PE-PDI seeds to the 2EH-PDI (50 μ M) dormant monomers in MCH* (a) at 0 seconds and (b) after 150 seconds.

Other minor points:

- ⇒ 1. “From FE-SEM studies, we found that the average length of the fibres prepared from [2EH-PDI_{monomer}] / [2EH-PDI_{seed}] ratios of 19:1, 3:1, and 1:1 are 6 μ m, 2 μ m and 1 μ m, respectively (Figure 4d-f).”

Comment: Seed size should be mentioned in this discussion

Ans. We appreciate this comment from the reviewer. In the revision, we have mentioned the size of the seed.

The following sentence is modified in the revised version of the manuscript. Please see page 8 of the revised manuscript

“From FE-SEM studies, we found that the average length of the fibres prepared from [2EH-PDI_{monomer}] / [2EH-PDI_{seed}] ratios of 19:1, 3:1, and 1:1 are 6 μ m, 2 μ m and 1 μ m, respectively, with the average seed length being 700 to 800 nm (Figure 4d-f).”

- ⇒ We found the distant nucleation events during the supramolecular polymerization of dormant monomers through homo-seeding, self-seeding and hetero-seeding, resulting in the creation of elegant supramolecular architectures such as 3D spherulites and scarf-like SP heterostructures.

Comment: “distant nucleation events” should be changed to “distinct nucleation events”

Ans. We thank the reviewer for finding his mistake. This has now been corrected in the revised manuscript.

⇒2. “Fig. 11” in SI should be changed to “Fig. S11”.

Ans. We thank the reviewer for finding his mistake. This has now been corrected in the revised manuscript.

⇒3. References should be checked properly. For example, volume is not bold in ref. 17. Similarly, in ref. 20, volume is italic instead of bold.

Ans. We thank the reviewer for finding his mistake. This has now been corrected in the revised manuscript.

⇒4. There are many formatting problems in the SI file. These should be checked properly.

Ans. We thank the reviewer for this comment. This has now been taken care of in the revised manuscript.

⇒5. d-spacings of the peaks should be assigned in Fig. S16.

Ans. We thank the reviewer for this comment. We have mentioned the d-spacings of the peaks in the revised version of the manuscript.

The revised Figure S22 is given below.

Figure S23. (a) Thin film XRD pattern of **2EH-PDI** (red), **PE-PDI** (blue) and **SP** heterostructures (green) obtained via hetero seeding approach by adding 50 mol% **PE-PDI**

seeds to **2EH-PDI** dormant monomers (50 μM) in MCH*. (b) Schematic illustration of proposed molecular packing of **PE-PDI** molecules in the 2D platelets.

Reviewer 3:

⇒ I co-reviewed this manuscript with one of the reviewers who provided the listed reports.

Ans. We thank the reviewer for providing valuable suggestions to improve the quality of our manuscript further.

Reviewer 4:

⇒ In this work, authors reported the formation of 3D aggregates via living supramolecular polymerization. The design and experiments are helpful for the development of supramolecular chemistry. The reviewer suggested to accept this manuscript. However there are several important issues.

Ans. We are grateful to the reviewer for recognising the importance of our work positively and for recommending it for publication.

⇒ The main weak point of this manuscript lies in the absence of detailed information of macroscopic aggregates

1. Authors applied FE-SEM/TEM to confirm the existence of 3D structures. However, there is no structural information obtained from SEM/TEM. No connection between supramolecular polymerization and 3D aggregates was available.

Ans. We thank the reviewer for this comment. We agree with the reviewer that the top-view of FE-SEM and TEM images provided in the manuscript is not sufficient to characterize the 3D structures of spherulites. To better understand the 3D structures of spherulites, we have recorded the FE-SEM images of spherulites by tilting the substrate up to 70° to have the side view (Figure S20c). The side view of spherulites clearly shows their 3D structure, which is constructed from supramolecular polymerization of **2EH-PDI** via a secondary nucleation-elongation mechanism. In this case, supramolecular polymerization via a secondary nucleation-elongation process takes place without the addition of seed via the application of mechanical agitation such as stirring and pipetting. This process is crucial to synthesize the 3D structures of spherulites. Otherwise, 1D supramolecular polymers can only be observed like in the case of homo-seeding where primary nucleation elongation is taking place.

We have added the following sentences and Figure 20c in the revised manuscript.

"The 3D structure of spherulite was further confirmed by FE-SEM images that provide side view, which was obtained by tilting the substrate up to 70°(Figure S20c). "

Figure S20: Time dependent absorption spectral changes of **2EH-PDI** dormant monomer solution after repetitive pipetting. (b) FE-SEM image (top view) (c) FE-SEM image in twisted angle measurement (Side view, 70°) and (d) TEM image of **2EH-PDI** one hour after applying repetitive pipetting ($c = 50 \mu\text{M}$, $l = 10 \text{ mm}$, Solvent: MCH* at 303K).

⇒2. The characterization of macroscopic aggregates was obviously insufficient. Authors did not provide any evidence of the chemical and physical properties of aggregates.

Ans. We thank the reviewer for this comment, which helped us further improve the novelty of our work. We have explored the differences in the physical, chemical and macroscopic properties of the SP architectures of **2EH-PDI** reported in the manuscript. The macroscopic appearance of solutions of 1D SPs prepared via primary nucleation is different than the 3D spherulites obtained via secondary nucleation. Due to the large 3D structure, the solutions containing spherulites appear inhomogeneous, whereas the solutions of 1D SPs are visibly homogeneous (Figure S16a). Next, we explored their thermal stability in MCH*. Interestingly, we found that the melting temperature of 3D spherulites is 7 K higher than 1D SPs (Figure S16b). We have also explored the chemical stability of these structures in 70% DCE in the

MCH mixture at 303 K (Figure S16c-e). We observed a quicker dissolution of 1D SPs of **2EH-PDI** into monomers than its 3D spherulites. These observations clearly indicate that the 3D spherulites of **2EH-PDI** formed via the secondary nucleation event are more robust than 1D SPs formed through the primary nucleation event. Moreover, 1D SPs of **2EH-PDI** are less thermally stable than **PE-PDI** in MCH*. Using these differences in thermal stability, we could selectively remove and reattach the 1D SPs from SP hetero architectures (Figures S26 and S27).

The following paragraphs and Figure are added in the revised manuscript. Please see page 11, first paragraph and page 15, bottom paragraph of the revised manuscript.

"Owing to their large 3D structures, the solutions having spherulites are inhomogeneous compared to the solutions having 1D SPs formed via primary nucleation (Figure S16a). We have also noticed that spherulites have better thermal stability than 1D SPs. A 50 μ M solution of **2EH-PDI** SPs with spherulite morphology synthesized via secondary nucleation (stirring at 60 RPM) showed a melting temperature of 312 K, which is 7 K higher than the 1D SPs obtained via primary nucleation (with 5 mol% seed) (Figure S16b). Moreover, when taken in 70% DCE in MCH at 303 K, 1D SPs are dissociated into monomers in less than 10 minutes, whereas 3D spherulites took more than 20 minutes (Figure S16c-e). All these observations suggest that the 3D spherulite structures of **2EH-PDI** formed via a secondary nucleation event are more robust than its 1D SPs formed via a primary nucleation event."

Figure S16: (a) Photographs of solutions containing 1D fibres (left) and 3D spherulites (right) of **2EH-PDI**. (b) Melting analysis of 1D fibres (blue) and 3D spherulites (green) of **2EH-PDI** monitored at 515 nm (heating rate 1K/min.). (c) Time-dependent supramolecular depolymerization of dried 1D fibres and 3D spherulites of **2EH-PDI** in 10 mm cell, after adding 70% DCE in MCH, monitored at 515 nm wavelength. (d) and (e) Time-dependent absorbance spectra of dried **2EH-PDI** 1D fibres solution (d) and dried 3D spherulites solution (e) in 10 mm cell after adding 70% DCE in MCH. ($c = 50 \mu\text{M}$, $l = 10 \text{ mm}$, Solvent: MCH*).

"Next, we have also explored the possibility of selectively disassembling one of the components from scarf-like 2D heterostructure. Interestingly, we found that SPs of **2EH-PDI** have less thermal stability than **PE-PDI** 2D platelets in MCH*. The SPs **2EH-PDI** showed melting at 307 K and significant depolymerization was observed at 323 K (Figure S26a,b). In contrast, **PE-PDI** seed particles are stable up to 330 K in MCH* (Figure S26a,c). Taking advantage of differences in thermal stability, the SP heterostructure solution was heated at 323 K for 15 minutes, and this hot solution was coated on the silicon wafer. FE-SEM images revealed the absence of SP heterostructure and the presence of segregated SPs of **2EH-PDI** and 2D platelets of **PE-PDI** (Figure S26d). This indicates that at 323 K, SPs of **2EH-PDI** are melted and detached from the surface of 2D platelets of **PE-PDI** and formed separately, probably during solvent evaporation. On the other hand, when this solution was cooled from 323 K to 303 K, and coated on a silicon wafer after waiting for an hour at 303 K, we observed the formation of SP heterostructures from FE-SEM images (Figure S26e,f). Furthermore, we immersed the silicon wafer containing SP heterostructures in MCH* solvent at 323 K for various time intervals and visualized them through FE-SEM, which showed only 2D platelet morphology (Figure S27b-d). This is due to the selective depolymerization and dissolution of SPs of **2EH-PDI** at 323 K from the surface of **PE-PDI** seeds into MCH* solution."

Figure S26: (a) Melting analysis of **2EH-PDI** 1D fibers (50 μM , MCH*) and **PE-PDI** seed particles prepared by sonicating for 1 hour (25 μM , MCH*). (b) UV-vis spectra of **2EH-PDI** 1D fibres (50 μM , MCH*) at 303 K (blue) and 323 K (red). (c) UV-vis spectra of **PE-PDI** seed particles (25 μM , MCH*) at 303 K (blue) and 323 K (red). (d), (e) and (f) Morphological characterization SP heterostructures at various temperatures by FE-SEM imaging. (d) FE-SEM image of sample collected from the solution of SP heterostructures heated to 323 K for 15 minutes. (e) and (f) FE-SEM images showing regeneration of SP hetero structures formed after cooling down the solution from 323 K to 303 and waited for 1 hour.

Figure S27: Morphological characterization of film-stated SP hetero structures at 323 K by time-dependent FE-SEM studies. (a) FE-SEM image of obtained SP hetero structures via hetero seeding approach. FE-SEM images of (b), (c) and (d) depict the removal of **2EH-PDI** fibers from SP heterostructures in MCH* at 323 K at various time intervals.

REVIEWERS' COMMENTS

Reviewer #1 (Remarks to the Author):

The authors were fortunate to receive thoughtful and detailed reviews. As Reviewer 1, I am satisfied that the revised manuscript addressed all of my concerns. However, I am not in a position to assess the responses to the comments of the other reviewers. If the other reviewers are also satisfied, then the manuscript should be accepted.

Reviewer #2 (Remarks to the Author):

The authors have substantially revised the experimental results and manuscript according to the reviewers' comments. All of the newly presented data make sense and warrant publication of this study in NatCommun.

Reviewer #3 (Remarks to the Author):

I co-reviewed this manuscript with one of the reviewers who provided the listed reports.

Reviewer #4 (Remarks to the Author):

Authors have carefully revised the maintext and SI according to the comments from the reviewer. The revised manuscript is ready for acceptance.